# Learning GANs and Ensembles Using Discrepancy

**Ben Adlam**
Google Research
New York, NY 10011
adlam@google.com

**Corinna Cortes**
Google Research
New York, NY 10011
corinna@google.com

**Mehryar Mohri**
Google Research & CIMS
New York, NY 10012
mohri@google.com

**Ningshan Zhang**
New York University
New York, NY 10012
nzhang@stern.nyu.edu

## Abstract

Generative adversarial networks (GANs) generate data based on minimizing a divergence between two distributions. The choice of that divergence is therefore critical. We argue that the divergence must take into account the hypothesis set and the loss function used in a subsequent learning task, where the data generated by a GAN serves for training. Taking that structural information into account is also important to derive generalization guarantees. Thus, we propose to use the *discrepancy* measure, which was originally introduced for the closely related problem of domain adaptation and which precisely takes into account the hypothesis set and the loss function. We show that discrepancy admits favorable properties for training GANs and prove explicit generalization guarantees. We present efficient algorithms using discrepancy for two tasks: training a GAN directly, namely DGAN, and mixing previously trained generative models, namely EDGAN. Our experiments on toy examples and several benchmark datasets show that DGAN is competitive with other GANs and that EDGAN outperforms existing GAN ensembles, such as AdaGAN.

## 1 Introduction

Generative adversarial networks (GANs) consist of a family of methods for unsupervised learning. A GAN learns a generative model that can easily output samples following a distribution $\mathbb{P}_\theta$, which aims to mimic the real data distribution $\mathbb{P}_r$. The parameter $\theta$ of the generator is learned by minimizing a divergence between $\mathbb{P}_r$ and $\mathbb{P}_\theta$, and different choices of this divergence lead to different GAN algorithms: the Jensen-Shannon divergence gives the standard GAN [Goodfellow et al., 2014, Salimans et al., 2016], the Wasserstein distance gives the WGAN [Arjovsky et al., 2017, Gulrajani et al., 2017], the squared maximum mean discrepancy gives the MMD GAN [Li et al., 2015, Dziugaite et al., 2015, Li et al., 2017], and the $f$-divergence gives the $f$-GAN [Nowozin et al., 2016], just to name a few. There are many other GANs that have been derived using other divergences in the past, see [Goodfellow, 2017] and [Creswell et al., 2018] for more extensive studies.

The choice of the divergence seems to be critical in the design of a GAN. But, how should that divergence be selected or defined? We argue that its choice must take into consideration the structure of a learning task and include, in particular, the hypothesis set and the loss function considered. In contrast, divergences that ignore the hypothesis set typically cannot benefit from any generalization guarantee (see for example Arora et al. [2017]). The loss function is also crucial: while many GAN applications aim to generate synthetic samples indistinguishable from original ones, for example images [Karras et al., 2018, Brock et al., 2019] or Anime characters [Jin et al., 2017], in many other

applications, the generated samples are used to improve subsequent learning tasks, such as data augmentation [Frid-Adar et al., 2018], improved anomaly detection [Zenati et al., 2018], or model compression [Liu et al., 2018b]. Such subsequent learning tasks require optimizing a specific loss function applied to the data. Thus, it would seem beneficial to explicitly incorporate this loss in the training of a GAN.

A natural divergence that accounts for both the loss function and the hypothesis set is the *discrepancy* measure introduced by Mansour et al. [2009]. Discrepancy plays a key role in the analysis of domain adaptation, which is closely related to the GAN problem, and other related problems such as drifting and time series prediction [Mohri and Medina, 2012, Kuznetsov and Mohri, 2015]. Several important generalization bounds for domain adaptation are expressed in terms of discrepancy [Mansour et al., 2009, Cortes and Mohri, 2014, Ben-David et al., 2007]. We define discrepancy in Section 2 and give examples illustrating the benefit of using discrepancy to measure the divergence between distributions.

In this work, we design a new GAN technique, *discrepancy GAN* (DGAN), that minimizes the discrepancy between $\mathbb{P}_\theta$ and $\mathbb{P}_r$. By training GANs with discrepancy, we obtain theoretical guarantees for subsequent learning tasks using the samples it generates. We show that discrepancy is continuous with respect to the generator's parameter $\theta$, under mild conditions, which makes training DGAN easy. Another key property of the discrepancy is that it can be accurately estimated from finite samples when the hypothesis set admits bounded complexity. This property does not hold for popular metrics such as the Jensen-Shannon divergence and the Wasserstein distance.

Moreover, we propose to use discrepancy to learn an ensemble of pre-trained GANs, which results in our EDGAN algorithm. By considering an ensemble of GANs, one can greatly reduce the problem of missing modes that frequently occurs when training a single GAN. We show that the discrepancy between the true and the ensemble distribution learned on finite samples converges to the discrepancy between the true and the optimal ensemble distribution, as the sample size increases. We also show that the EDGAN problem can be formulated as a convex optimization problem, thereby benefiting from strong convergence guarantees. Recent work of Tolstikhin et al. [2017], Arora et al. [2017], Ghosh et al. [2018] and Hoang et al. [2018] also considered mixing GANs, either motived by boosting algorithms such as AdaBoost, or by the minimax theorem in game theory. These algorithms train multiple generators and learn the mixture weights simultaneously, yet none of them explicitly optimizes for the mixture weights once the multiple GANs are learned, which can provide additional improvement as demonstrated by our experiments with EDGAN.

The term "discrepancy" has been previously used in the GAN literature under a different definition. The *squared maximum mean discrepancy (MMD)*, which was originally proposed by Gretton et al. [2012], is used as the distance metric for training MMD GAN [Li et al., 2015, Dziugaite et al., 2015, Li et al., 2017]. MMD between two distributions is defined with respect to a family of functions $\mathcal{F}$, which is usually assumed to be a reproducing kernel Hilbert space (RKHS) induced by a kernel function, but MMD does not take into account the loss function. LSGAN [Mao et al., 2017] also adopts the squared loss function for the discriminator, and as we do for DGAN. Feizi et al. [2017], Deshpande et al. [2018] consider minimizing the quadratic Wasserstein distance between the true and the generated samples, which involves the squared loss function as well. However, their training objectives are vastly different from ours. Finally, when the hypothesis set is the family of linear functions with bounded norm and the loss function is the squared loss, DGAN coincides with the objective sought by McGAN [Mroueh et al., 2017], that of matching the empirical covariance matrices of the true and the generated distribution. However, McGAN uses nuclear norm while DGAN uses spectral norm in that case.

The rest of this paper is organized as follows. In Section 2, we define discrepancy and prove that it benefits from several favorable properties, including continuity with respect to the generator's parameter and the possibility of accurately estimating it from finite samples. In Section 3, we describe our discrepancy GAN (DGAN) and ensemble discrepancy GAN (EDGAN) algorithms with a discussion of the optimization solution and theoretical guarantees. We report the results of a series of experiments (Section 4), on both toy examples and several benchmark datasets, showing that DGAN is competitive with other GANs and that EDGAN outperforms existing GAN ensembles, such as AdaGAN.

## 2  Discrepancy

Let $\mathbb{P}_r$ denote the real data distribution on $\mathcal{X}$, which, without loss of generality, we can assume to be $\mathcal{X} = \{x \in \mathbb{R}^d \colon \|x\|_2 \leq 1\}$. A GAN generates a sample in $\mathcal{X}$ via the following procedure: it first draws a random noise vector $z \in \mathcal{Z}$ from a fixed distribution $\mathbb{P}_z$, typically a multivariate Gaussian, and then passes $z$ through the generator $g_\theta \colon \mathcal{Z} \to \mathcal{X}$, typically a neural network parametrized by $\theta \in \Theta$. Let $\mathbb{P}_\theta$ denote the resulting distribution of $g_\theta(z)$. Given a distance metric $d(\cdot, \cdot)$ between two distributions, a GAN's learning objective is to minimize $d(\mathbb{P}_r, \mathbb{P}_\theta)$ over $\theta \in \Theta$.

In Appendix A, we present and discuss two instances of the distance metric $d(\cdot, \cdot)$ and two widely-used GANs: the Jensen-Shannon divergence for the standard GAN [Goodfellow et al., 2014], and the Wasserstein distance for WGAN [Arjovsky et al., 2017]. Furthermore, we show that Wasserstein distance can be viewed as discrepancy without considering the hypothesis set and the loss function, which is one of the reasons why it cannot benefit from theoretical guarantees. In this section, we describe the discrepancy measure and motivate its use by showing that it benefits from several important favorable properties.

Consider a hypothesis set $\mathcal{H}$ and a symmetric loss function $\ell \colon \mathcal{Y} \times \mathcal{Y} \to \mathbb{R}$, which will be used in future supervised learning tasks on the true (and probably also the generated) data. Given $\mathcal{H}$ and $\ell$, the discrepancy between two distributions $\mathbb{P}$ and $\mathbb{Q}$ is defined by the following:

$$\text{disc}_{\mathcal{H},\ell}(\mathbb{P}, \mathbb{Q}) = \sup_{h,h' \in \mathcal{H}} \Big| \mathop{\mathbb{E}}_{x \sim \mathbb{P}} \big[ \ell\big(h(x), h'(x)\big) \big] - \mathop{\mathbb{E}}_{x \sim \mathbb{Q}} \big[ \ell\big(h(x), h'(x)\big) \big] \Big|. \tag{1}$$

Equivalently, let $\ell_{\mathcal{H}} = \big\{ \ell\big(h(x), h'(x)\big) \colon h, h' \in \mathcal{H} \big\}$ be the family of discriminators induced by $\ell$ and $\mathcal{H}$, then, the discrepancy can be written as $\text{disc}_{\mathcal{H},\ell}(\mathbb{P}, \mathbb{Q}) = \sup_{f \in \ell_{\mathcal{H}}} \big| \mathbb{E}_{\mathbb{P}}[f(x)] - \mathbb{E}_{\mathbb{Q}}[f(x)] \big|$.

How would subsequent learning tasks benefit from samples generated by GANs trained with discrepancy? We show that, under mild conditions, any hypothesis performing well on $\mathbb{P}_\theta$ (with loss function $\ell$) is guaranteed to perform well on $\mathbb{P}_r$, as long as the discrepancy $\text{disc}_{\mathcal{H},\ell}(\mathbb{P}_\theta, \mathbb{P}_r)$ is small.

**Theorem 1.** *Assume the true labeling function $f \colon \mathcal{X} \to \mathcal{Y}$ is contained in the hypothesis set $\mathcal{H}$. Then, for any hypothesis $h \in \mathcal{H}$,*

$$\mathop{\mathbb{E}}_{x \sim \mathbb{P}_r} [\ell(h, f)] \leq \mathop{\mathbb{E}}_{x \sim \mathbb{P}_\theta} [\ell(h, f)] + \text{disc}_{\mathcal{H},\ell}(\mathbb{P}_\theta, \mathbb{P}_r).$$

Theorem 1 suggests that the learner can learn a model using samples drawn from $\mathbb{P}_\theta$, whose generation error on $\mathbb{P}_r$ is guaranteed to be no more than its generation error on $\mathbb{P}_\theta$ plus the discrepancy, which is minimized by the algorithm. The proof uses the definition of discrepancy. Due to space limitation, we provide all the proofs in Appendix B.

### 2.1  Hypothesis set and loss function

We argue that discrepancy is more favorable than Wasserstein distance measures, since it makes explicit the dependence on loss function and hypothesis set. We consider two widely used learning scenarios: 0-1 loss with linear separators, and squared loss with Lipschitz functions.

**0-1 Loss, Linear Separators**  Consider the two distributions on $\mathbb{R}^2$ illustrated in Figure 1a: $\mathbb{Q}$ (filled circles ●) is equivalent to $\mathbb{P}$ (circles ○), but with all points shifted to the right by a small amount $\epsilon$. Then, by the definition of Wasserstein distance, $W(\mathbb{P}, \mathbb{Q}) = \epsilon$, since to transport $\mathbb{P}$ to $\mathbb{Q}$, one just need to move each point to the right by $\epsilon$. When $\epsilon$ is small, WGAN views the two distributions as close and thus stops training. On the other hand, when $\ell$ is the 0-1 loss and $\mathcal{H}$ is the set of linear separators, $\text{disc}_{\mathcal{H},\ell}(\mathbb{P}, \mathbb{Q}) = 1$, which is achieved at the $h, h'$ as shown in Figure 1a, with $\mathbb{E}_{\mathbb{P}}[1_{h(x) \neq h'(x)}] = 1$ and $\mathbb{E}_{\mathbb{Q}}[1_{h(x) \neq h'(x)}] = 0$. Thus, DGAN continues training to push $\mathbb{Q}$ towards $\mathbb{P}$.

The example above is an extreme case where $\mathbb{P}$ and $\mathbb{Q}$ are separable. In more practical scenarios, the domain of the two distributions may overlap significantly, as illustrated in Figure 1b, where $\mathbb{P}$ is in red and $\mathbb{Q}$ is in blue, and the shaded areas contain 95% probably mass. Again, $\mathbb{Q}$ equals $\mathbb{P}$ shifting to the right by $\epsilon$ and thus $W(\mathbb{P}, \mathbb{Q}) = \epsilon$. Since the non-overlapping area has a sizable probability mass, the discrepancy between $\mathbb{P}$ and $\mathbb{Q}$ is still large, for the same reason as for Figure 1a.

These examples demonstrate the importance of taking hypothesis sets and loss functions into account when comparing two distributions: even though two distributions appear geometrically "close"

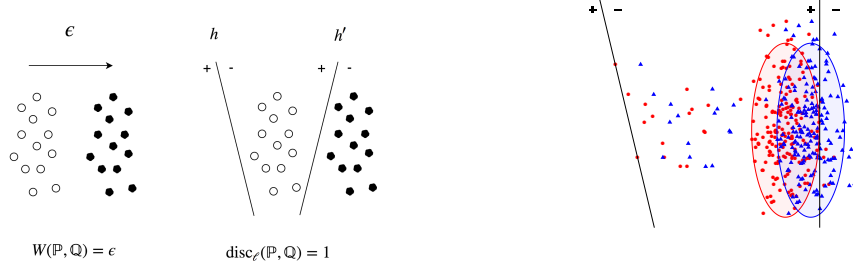

(a) Non-overlapping distributions, $\mathbb{P}$: $\{\circ\}$, $\mathbb{Q}$: $\{\bullet\}$.　　(b) Overlapping distributions, $\mathbb{P}$: {red}, $\mathbb{Q}$: {blue}.

Figure 1: Distributions $\mathbb{P}$ and $\mathbb{Q}$ may appear "close" under Wasserstein distance, but the discrepancy between the two is still large, where the discrepancy is defined by 0-1 loss and linear separators.

under Wasserstein distance, a classifier trained on one distribution may perform poorly on another distribution. According to Theorem 1, such unfortunate behaviors are less likely to happen with $\mathrm{disc}_{\mathcal{H},\ell}$.

**Squared Loss, Lipschitz Functions**　　Next, we consider the squared loss and the hypothesis set of 1-Lipschitz functions $\mathcal{H} = \{h\colon |h(x) - h(x')| \le \|x - x'\|_2, \forall x, x' \in \mathcal{X}\}$, then $\ell_{\mathcal{H}} = \{[h(x) - h'(x)]^2\colon h, h' \in \mathcal{H}\}$. We can show that $\ell_{\mathcal{H}}$ is a subset of 4-Lipschitz functions on $\mathcal{X}$. Then, by the definition of discrepancy and Wasserstein distance, $\mathrm{disc}_{\mathcal{H},\ell}(\mathbb{P}, \mathbb{Q})$ is comparable to $W(\mathbb{P}, \mathbb{Q})$:

$$\mathrm{disc}_{\mathcal{H},\ell}(\mathbb{P}, \mathbb{Q}) = \sup_{f \in \ell_{\mathcal{H}}} \mathop{\mathbb{E}}_{\mathbb{P}}\big[f(x)\big] - \mathop{\mathbb{E}}_{\mathbb{Q}}\big[f(x)\big] \le \sup_{f\colon \text{4-Lipschitz}} \mathop{\mathbb{E}}_{\mathbb{P}}\big[f(x)\big] - \mathop{\mathbb{E}}_{\mathbb{Q}}\big[f(x)\big] = 4W(\mathbb{P}, \mathbb{Q}).$$

However, the inequality above can be quite loose since, depending on the hypothesis set, $\ell_{\mathcal{H}}$ may be only a small subset of all 4-Lipschitz functions. For instance, when $\mathcal{H}$ is the set of linear functions with norm bounded by one, then $\ell_{\mathcal{H}} = \{(w^T x)^2\colon \|w\| \le 2\}$, which is a significantly smaller set than the family of all 4-Lipschitz functions. Thus, $\mathrm{disc}_{\mathcal{H},\ell}(\mathbb{P}, \mathbb{Q})$ could potentially be a tighter measure than $W(\mathbb{P}, \mathbb{Q})$, depending on $\mathcal{H}$.

## 2.2　Continuity and estimation

In this section, we discuss two favorable properties of discrepancy: its continuity under mild assumptions with respect to the generator's parameter $\theta$, a property shared with the Wasserstein distance, and the fact that it can be accurately estimated from finite samples, which does not hold for either the Jensen-Shannon or the Wasserstein distance. The continuity property is summarized in the following theorem.

**Theorem 2.** *Let $\mathcal{H} = \{h\colon \mathcal{X} \to \mathcal{Y}\}$ be a family of $\mu$-Lipschitz functions and assume that the loss function $\ell$ is continuous and symmetric in its arguments, and is bounded by $M$. Assume further that $\ell$ admits the triangle inequality, or that it can be written as $\ell(y, y') = f(|y - y'|)$ for some Lipschitz function $f$. Assume that $g_\theta\colon \mathcal{Z} \to \mathcal{X}$ is continuous in $\theta$. Then, $\mathrm{disc}_{\mathcal{H},\ell}(\mathbb{P}_r, \mathbb{P}_\theta)$ is continuous in $\theta$.*

The assumptions of Theorem 2 are easily satisfied in practice, where $h \in \mathcal{H}$ and $g_\theta$ are neural networks whose parameters are limited within a compact set, and where the loss function can be either the $\ell_1$ loss, $\ell(y, y') = |y - y'|$, or the squared loss, $\ell(y, y') = (y - y')^2$. If the discrepancy is continuous in $\theta$, then, as the sequence of parameters $\theta_t$ converges to $\theta^*$, the discrepancy also converges: $|\mathrm{disc}_{\mathcal{H},\ell}(\mathbb{P}_r, \mathbb{P}_{\theta_t}) - \mathrm{disc}_{\mathcal{H},\ell}(\mathbb{P}_r, \mathbb{P}_{\theta^*})| \to 0$, which is a desirable property for training DGAN. The reader is referred to Arjovsky et al. [2017] for a more extensive discussion of the continuity properties of various distance metrics and their effects on training GANs.

Next, we show that discrepancy can be accurately estimated from finite samples. Let $S_r$ and $S_\theta$ be i.i.d. samples drawn from $\mathbb{P}_r$ and $\mathbb{P}_\theta$ with $|S_r| = m$ and $|S_\theta| = n$, and let $\widehat{\mathbb{P}}_r$ and $\widehat{\mathbb{P}}_\theta$ be the empirical distributions induced by $S_r$ and $S_\theta$, respectively. Recall that the empirical Radmacher complexity of a hypothesis set $\mathcal{G}$ on sample $S$ of size $m$ is defined by: $\widehat{\mathfrak{R}}_S(\mathcal{G}) = \frac{2}{m} \mathbb{E}_\sigma \big[\sup_{g \in \mathcal{G}} \sum_{i=1}^m \sigma_i g(x_i)\big]$, where $\sigma_1, \sigma_2, \ldots, \sigma_m$ are i.i.d. random variables with $\mathbb{P}(\sigma_i = 1) = \mathbb{P}(\sigma_i = -1) = 1/2$. The empirical Radmacher complexity measures the complexity of the hypothesis set $\mathcal{G}$. The next theorem presents the learning guarantees of discrepancy.

**Theorem 3.** *Assume the loss is bounded, $\ell \leq M$. For any $\delta > 0$, with probability at least $1 - \delta$ over the draw of $S_r$ and $S_\theta$,*

$$\left| disc_{\mathcal{H},\ell}(\mathbb{P}_r, \mathbb{P}_\theta) - disc_{\mathcal{H},\ell}(\widehat{\mathbb{P}}_r, \widehat{\mathbb{P}}_\theta) \right| \leq \widehat{\mathfrak{R}}_{S_r}(\ell_{\mathcal{H}}) + \widehat{\mathfrak{R}}_{S_\theta}(\ell_{\mathcal{H}}) + 3M\left( \sqrt{\tfrac{\log(4/\delta)}{2m}} + \sqrt{\tfrac{\log(4/\delta)}{2n}} \right).$$

*Furthermore, when the loss function $\ell(h, h')$ is a q-Lipschitz function of $h - h'$, we have*

$$\left| disc_{\mathcal{H},\ell}(\mathbb{P}_r, \mathbb{P}_\theta) - disc_{\mathcal{H},\ell}(\widehat{\mathbb{P}}_r, \widehat{\mathbb{P}}_\theta) \right| \leq 4q\left( \widehat{\mathfrak{R}}_{S_r}(\mathcal{H}) + \widehat{\mathfrak{R}}_{S_\theta}(\mathcal{H}) \right) + 3M\left( \sqrt{\tfrac{\log(4/\delta)}{2m}} + \sqrt{\tfrac{\log(4/\delta)}{2n}} \right).$$

In the rest of this paper, we will consider the squared loss $\ell(y, y') = (y - y')^2$, which is bounded and 2-Lipschitz when $|h(x)| \leq 1$ for all $h \in \mathcal{H}$ and $x \in \mathcal{X}$. Furthermore, when $\mathcal{H}$ is a family of feedforward neural networks, Cortes et al. [2017] provided an explicit upper bound of $\widehat{\mathfrak{R}}_S(\mathcal{H}) = O(1/\sqrt{m})$ for its complexity, and thus the right-hand side of the above inequality is in $O(\frac{1}{\sqrt{m}} + \frac{1}{\sqrt{n}})$. Then, for $m$ and $n$ sufficiently large, the empirical discrepancy is close to the true discrepancy. It is important that the discrepancy can be accurately estimated from finite samples since, when training DGAN, we can only approximate the true discrepancy with a batch of samples. In contrast, the Jensen-Shannon distance and the Wasserstein distance do not admit this favorable property [Arora et al., 2017].

## 3  Algorithms

In this section, we show how to compute the discrepancy and train DGAN for various hypothesis sets and the squared loss. We also propose to learn an ensemble of pre-trained GANs via minimizing discrepancy. We name this method EDGAN, and present its learning guarantees.

### 3.1  DGAN algorithm

Given a parametric family of hypotheses $\mathcal{H} = \{h_w \colon w \in W\}$, DGAN is defined as the following min-max optimization problem:

$$\min_{\theta \in \Theta} \max_{w, w' \in W} \left| \mathop{\mathbb{E}}_{x \sim \mathbb{P}_r} \left[ \ell\big(h_w(x), h_{w'}(x)\big) \right] - \mathop{\mathbb{E}}_{x \sim \mathbb{P}_\theta} \left[ \ell\big(h_w(x), h_{w'}(x)\big) \right] \right|. \tag{2}$$

As with other GANs, DGAN is trained by iteratively solving the min-max problem (2). The minimization over the generator's parameters $\theta$ can be tackled by standard stochastic gradient descent (SGD) algorithm with back-propagation. The inner maximization problem that computes the discrepancy, however, can be efficiently solved when $\ell$ is the squared loss function.

We first consider $\mathcal{H}$ to be the set of linear functions with bounded norm: $\mathcal{H} = \{x \to w^T x \colon \|w\|_2 \leq 1, w \in \mathbb{R}^d\}$. Recall the definition of $S_r$, $S_\theta$, $\mathbb{P}_r$ and $\mathbb{P}_\theta$ from Section 2.2. In addition, let $X_r$ and $X_\theta$ denote the corresponding $m \times d$ and $n \times d$ data matrices, where each row represents one input.

**Proposition 4.** *When $\ell$ is the squared loss and $\mathcal{H}$ the family of linear functions with norm bounded by 1, $disc_{\mathcal{H},\ell}(\widehat{\mathbb{P}}_r, \widehat{\mathbb{P}}_\theta) = 2 \left\| \frac{1}{n} X_\theta^T X_\theta - \frac{1}{m} X_r^T X_r \right\|_2$, where $\| \cdot \|_2$ denotes the spectral norm.*

Thus, the discrepancy $disc_{\mathcal{H},\ell}(\widehat{\mathbb{P}}_r, \widehat{\mathbb{P}}_\theta)$ equals twice the largest eigenvalue in absolute value of the data-dependent matrix $\boldsymbol{M}(\theta) = \frac{1}{n} X_\theta^T X_\theta - \frac{1}{m} X_r^T X_r$. Given $v^*(\theta)$, the corresponding eigenvector at the optimal solution, we can then back-propagate the loss $disc_{\mathcal{H},\ell}(\widehat{\mathbb{P}}_r, \widehat{\mathbb{P}}_\theta) = 2v^{*T}(\theta)\boldsymbol{M}(\theta)v^*(\theta)$ to optimize for $\theta$. The maximum or minimum eigenvalue of $\boldsymbol{M}(\theta)$ can be computed in $O(d^2)$ [Golub and van Van Loan, 1996], and the power method can be used to closely approximate it.

The closed-form solution in Proposition 4 holds for a family $\mathcal{H}$ of linear mappings. To generate realistic outcomes with DGAN, however, we need a more complex hypothesis set $\mathcal{H}$, such as the family of deep neural networks (DNN). Thus, we consider the following approach: first, we fix a pre-trained DNN classifier, such as the inception network, and pass the samples through this network to obtain the last (or any other) layer of embedding $f \colon \mathcal{X} \to \mathcal{E}$, where $\mathcal{E}$ is the embedding space. Next, we compute the discrepancy on the embedded samples with $\mathcal{H}$ being the family of linear functions with bounded norm, which admits a closed-form solution according to Proposition 4. In practice, it also makes sense to train the embedding network together with the generator: let $f_\zeta$ be the embedding network parametrized by $\zeta$, then DGAN optimizes for both $f_\zeta$ and $g_\theta$. See Algorithm 1 for a single step of updating DGAN. In particular, the learner can either compute $F(\zeta^t, \theta^t)$ exactly, or use an approximation based on the power method. Note that when the learner uses a pre-fixed embedding network $f$, the update step of $\zeta^{t+1}$ can be skipped.

| **Algorithm 1** UPDATE DGAN($\zeta^t, \theta^t, \eta$) | **Algorithm 2** UPDATE EDGAN($\boldsymbol{\alpha}^t, f, \eta$) |
|---|---|
| $X_r \leftarrow [f_{\zeta^t}(x_1), \cdots, f_{\zeta^t}(x_m)]^T$, where $x_i \sim \mathbb{P}_r$ | $X_r \leftarrow [f(x_1), \cdots, f(x_{n_r})]^T$, where $x_i \sim \mathbb{P}_r$ |
| $X_\theta \leftarrow [f_{\zeta^t}(x'_1), \cdots, f_{\zeta^t}(x'_n)]^T$, where $x'_i \sim \mathbb{P}_{\theta^t}$ | $X_k \leftarrow [f(x_1^k), \cdots, f(x_{n_k}^k)]^T$, where $x_i^k \sim \mathbb{P}_{\theta_k}$ |
| $F(\zeta^t, \theta^t) \leftarrow \left\| \frac{1}{n} X_\theta^T X_\theta - \frac{1}{m} X_r^T X_r \right\|_2$ | $F(\boldsymbol{\alpha}^t) \leftarrow \| \left( \sum_{k=1}^p \frac{\alpha_k^t}{n_k} X_k^T X_k \right) - \frac{1}{n_r} X_r^T X_r \|_2$ |
| Update: $\zeta^{t+1} \leftarrow \zeta^t + \eta \nabla_\zeta F(\zeta^t, \theta^t)$ | Update: $\boldsymbol{\alpha}^{t+1} \leftarrow \boldsymbol{\alpha}^t - \eta \nabla_{\boldsymbol{\alpha}} F(\boldsymbol{\alpha}^t)$ |
| Update: $\theta^{t+1} \leftarrow \theta^t - \eta \nabla_\theta F(\zeta^t, \theta^t)$ | |

## 3.2   EDGAN algorithm

Next, we show that discrepancy provides a principled way of choosing the ensemble weights to mix pre-trained GANs, which admits favorable convergence guarantees.

Let $g_1, \ldots, g_p$ be $p$ pre-trained GANs. For a given mixture weight $\boldsymbol{\alpha} = (\alpha_1, \ldots, \alpha_p) \in \Delta$, where $\Delta = \{(\alpha_1, \ldots, \alpha_p) \colon \alpha_k \geq 0, \sum_{k=1}^p \alpha_k = 1\}$ is the simplex in $\mathbb{R}^p$, we define the ensemble of $p$ GANs by $g_{\boldsymbol{\alpha}} = \sum_{k=1}^p \alpha_k g_k$. To draw a sample from the ensemble $g_{\boldsymbol{\alpha}}$, we first sample an index $k \in [p] = \{1, 2, \cdots, p\}$ according to the multinomial distribution with parameter $\boldsymbol{\alpha}$, and then return a random sample generated by the chosen GAN $g_k$. We denote by $\mathbb{P}_{\boldsymbol{\alpha}}$ the distribution of $g_{\boldsymbol{\alpha}}$. EDGAN determines the mixture weight $\boldsymbol{\alpha}$ by minimizing the discrepancy between $\mathbb{P}_{\boldsymbol{\alpha}}$ and the real data $\mathbb{P}_r$: $\min_{\boldsymbol{\alpha} \in \Delta} \text{disc}_{\mathcal{H}, \ell}(\mathbb{P}_{\boldsymbol{\alpha}}, \mathbb{P}_r)$.

To learn the mixture weight $\boldsymbol{\alpha}$, we approximate the true distributions by their empirical counterparts: for each $k \in [p]$, we randomly draw a set of $n_k$ samples from $g_k$, and randomly draw $n_r$ samples from the real data distribution $\mathbb{P}_r$. Let $S_k$ and $S_r$ denote the corresponding set of samples, and let $\widehat{\mathbb{P}}_k$ and $\widehat{\mathbb{P}}_r$ denote the induced empirical distributions, respectively. For a given $\boldsymbol{\alpha}$, let $\widehat{\mathbb{P}}_{\boldsymbol{\alpha}} = \sum_{k=1}^p \alpha_k \widehat{\mathbb{P}}_k$ be the empirical counterparts of $\mathbb{P}_{\boldsymbol{\alpha}}$. We first present a convergence result for the EDGAN method, and then describe how to train EDGAN.

Let $\boldsymbol{\alpha}^*$ and $\widehat{\boldsymbol{\alpha}}$ be the discrepancy minimizer under the true and the empirical distributions, respectively:

$$\boldsymbol{\alpha}^* = \operatorname*{argmin}_{\boldsymbol{\alpha} \in \Delta} \text{disc}_{\mathcal{H}, \ell}(\mathbb{P}_{\boldsymbol{\alpha}}, \mathbb{P}_r), \quad \widehat{\boldsymbol{\alpha}} = \operatorname*{argmin}_{\boldsymbol{\alpha} \in \Delta} \text{disc}_{\mathcal{H}, \ell}(\widehat{\mathbb{P}}_{\boldsymbol{\alpha}}, \widehat{\mathbb{P}}_r).$$

For simplicity, we set $n_k = n_r = n$ for all $k \in [p]$, but the following result can be easily extended to arbitrary batch size for each generator.

**Theorem 5.** *For any $\delta > 0$, with probability at least $1 - \delta$ over the draw of samples,*

$$|\text{disc}_{\mathcal{H}, \ell}(\mathbb{P}_{\widehat{\boldsymbol{\alpha}}}, \mathbb{P}_r) - \text{disc}_{\mathcal{H}, \ell}(\mathbb{P}_{\boldsymbol{\alpha}^*}, \mathbb{P}_r)| \leq 2\Big(\widehat{\mathfrak{R}}_S(\ell_{\mathcal{H}}) + 3M\sqrt{\log[4(p+1)/\delta]/2n}\Big),$$

*where $\widehat{\mathfrak{R}}_S(\ell_{\mathcal{H}}) = \max\big\{\widehat{\mathfrak{R}}_{S_1}(\ell_{\mathcal{H}}), \ldots, \widehat{\mathfrak{R}}_{S_p}(\ell_{\mathcal{H}}), \widehat{\mathfrak{R}}_{S_r}(\ell_{\mathcal{H}})\big\}$. Furthermore, when the loss function $\ell(h, h')$ is a $q$-Lipschitz function of $h - h'$, the following holds with probability $1 - \delta$:*

$$|\text{disc}_{\mathcal{H}, \ell}(\mathbb{P}_{\widehat{\boldsymbol{\alpha}}}, \mathbb{P}_r) - \text{disc}_{\mathcal{H}, \ell}(\mathbb{P}_{\boldsymbol{\alpha}^*}, \mathbb{P}_r)| \leq 2\Big(4q\widehat{\mathfrak{R}}_S(\mathcal{H}) + 3M\sqrt{\log[4(p+1)/\delta]/2n}\Big),$$

*where $\widehat{\mathfrak{R}}_S(\mathcal{H}) = \max\big\{\widehat{\mathfrak{R}}_{S_1}(\mathcal{H}), \ldots, \widehat{\mathfrak{R}}_{S_p}(\mathcal{H}), \widehat{\mathfrak{R}}_{S_r}(\mathcal{H})\big\}$.*

When $\ell$ is the squared loss and $\mathcal{H}$ is the family of feedforward neural networks, the upper bound on $\widehat{\mathfrak{R}}_S(\ell_{\mathcal{H}})$ is in $O(1/\sqrt{n})$. Since we can generate unlimited samples from each of the $p$ pre-trained GANs, $n$ can be as large as the number of available real samples, and thus the discrepancy between the learned ensemble $\mathbb{P}_{\widehat{\boldsymbol{\alpha}}}$ and the real data $\mathbb{P}_r$ can be very close to the discrepancy between the optimal ensemble $\mathbb{P}_{\boldsymbol{\alpha}^*}$ and the real data $\mathbb{P}_r$. This is a very favorable generalization guarantee for EDGAN, since it suggests that the mixture weight learned on the training data is guaranteed to generalize and perform well on the test data, a fact also corroborated by our experiments.

To compute the discrepancy for EDGAN, we again begin with linear mappings $\mathcal{H} = \{x \rightarrow w^T x \colon \|w\|_2 \leq 1, w \in \mathbb{R}^d\}$. For each generator $k \in [p]$, we obtain a $n_k \times d$ data matrix $X_k$, and similarly we have the $n_r \times d$ data matrix for the real samples. Then, by the proof of Proposition 4, discrepancy minimization can be written as

$$\min_{\boldsymbol{\alpha} \in \Delta} \text{disc}_{\mathcal{H}, \ell}(\widehat{\mathbb{P}}_{\boldsymbol{\alpha}}, \widehat{\mathbb{P}}_r) = 2\min_{\boldsymbol{\alpha} \in \Delta} \|\boldsymbol{M}(\boldsymbol{\alpha})\|_2, \text{ with } \boldsymbol{M}(\boldsymbol{\alpha}) = \left[\sum_{k=1}^p \frac{\alpha_k}{n_k} X_k^T X_k\right] - \frac{1}{n_r} X_r^T X_r. \quad (3)$$

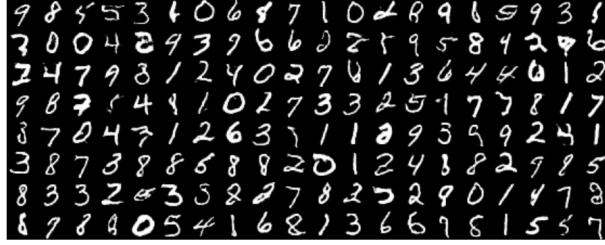
Figure 2: Random samples from DGAN trained on MNIST.

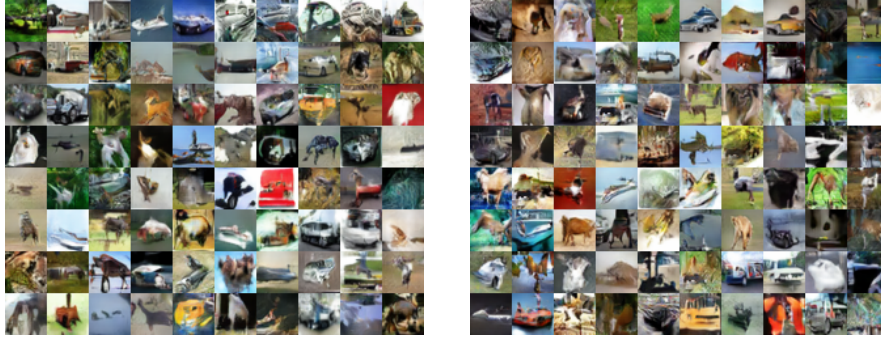
Figure 3: Random samples from DGAN trained on CIFAR10.

Since $\boldsymbol{M}(\boldsymbol{\alpha})$ and $-\boldsymbol{M}(\boldsymbol{\alpha})$ are affine and thus convex functions of $\boldsymbol{\alpha}$, $\|\boldsymbol{M}(\boldsymbol{\alpha})\|_2 = \sup_{\|v\|_2 \le 1} |v^T \boldsymbol{M}(\boldsymbol{\alpha})v|$ is also convex in $\boldsymbol{\alpha}$, as the supremum of a set of convex functions is convex. Thus, problem (3) is a convex optimization problem, thereby benefitting from strong convergence guarantees.

Note that we have $\|\boldsymbol{M}(\boldsymbol{\alpha})\|_2 = \max\{\lambda_{\max}(\boldsymbol{M}(\boldsymbol{\alpha})), \lambda_{\max}(-\boldsymbol{M}(\boldsymbol{\alpha}))\}$. Thus, one way to solve problem (3) is to cast it as a semi-definite programming (SDP) problem:

$$\min_{\boldsymbol{\alpha}, \lambda} \quad \lambda, \qquad \text{s.t.} \quad \lambda \boldsymbol{I} - \boldsymbol{M}(\boldsymbol{\alpha}) \succeq 0, \ \lambda \boldsymbol{I} + \boldsymbol{M}(\boldsymbol{\alpha}) \succeq 0, \ \boldsymbol{\alpha} \ge 0, \ \mathbf{1}^T \boldsymbol{\alpha} = 1.$$

An alternative solution consists of using the power method to approximate the spectral norm, which is faster when the sample dimension $d$ is large. As with DGAN, we can also consider a more complex hypothesis set $\mathcal{H}$, by first passing samples through an embedding network $f$, and then letting $\mathcal{H}$ be the set of linear mappings on the embedded samples. Since the generators are already pre-trained for EDGAN, we no longer need to train the embedding network, but instead keep it fixed. See Algorithm 2 for one training step of EDGAN.

## 4 Experiments

### 4.1 DGAN

In this section, we show that DGAN obtains competitive results on the benchmark datasets MNIST, CIFAR10, CIFAR100, and CelebA (at resolution $128 \times 128$). We did unconditional generation and did not use the labels in the dataset. We trained both the discriminator's embedding layer and the generator with discrepancy loss as in Algorithm 1. Note, we did not attempt to optimize the architecture and other hyperparameters to get state-of-the-art results. We used a standard DCGAN architecture. The main architectural modification for DGAN is that the final dense layer of the discriminator has output dimension greater than 1 since, in DGAN, the discriminator outputs an embedding layer rather than a single score. The size of this embedding layer is a hyperparameter that can be tuned, but we refrained from doing so here. See Table 6 in Appendix C for DGAN architectures. One important observation is that larger embedding layers require more samples to accurately estimate the population covariance matrix of the embedding layer under the data and generated distributions (and hence the spectral norm of the difference).

To enforce the Lipschitz assumption of our Theorems, either weight clipping [Arjovsky et al., 2017],

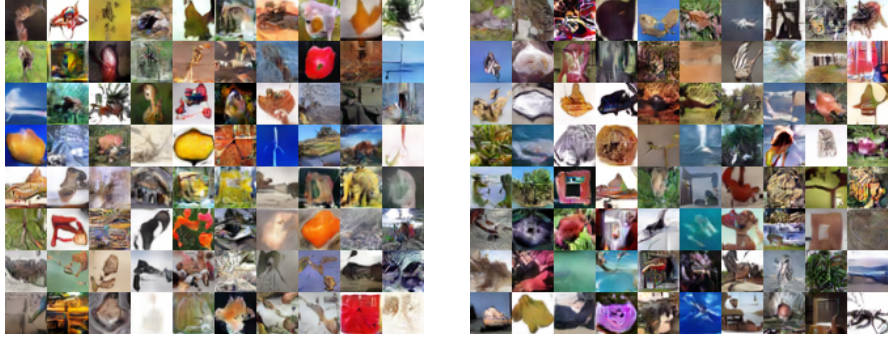

Figure 4: Random samples from DGAN trained on CIFAR100.

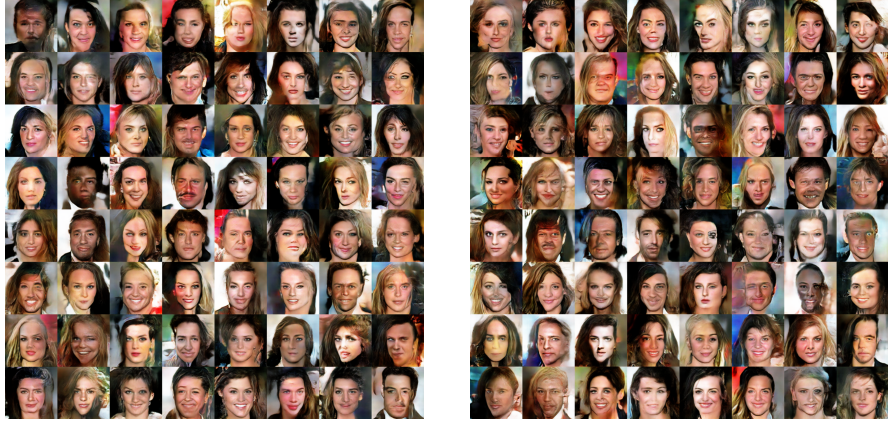

Figure 5: Random samples from DGAN trained on CelebA at resolution $128 \times 128$.

gradient penalization [Gulrajani et al., 2017], spectral normalization [Miyato et al., 2018], or some combination can be used. We found gradient penalization useful for its stabilizing effect on training, and obtained the best performance with this and weight clipping. Table 1 lists Inception score (IS) and Fréchet Inception distance (FID) on various datasets. All results are the best of five trials. While our scores are not state-of-the-art [Brock et al., 2019], they are close to those achieved by similar unconditional DCGANs [Miyato et al., 2018, Lucic et al., 2018]. Figures 2-5 show samples from a trained DGAN that are not cherry-picked.

Table 1: Inception Score (IS) and Fréchet Inception Distance (FID) for various datasets.

| Dataset | IS | FID (train) | FID (test) |
|---------|------|-------------|------------|
| CIFAR10 | 7.02 | 26.7 | 30.7 |
| CIFAR100 | 7.31 | 28.9 | 33.3 |
| CelebA | 2.15 | 59.2 | - |

## 4.2 EDGAN

**Toy example**   We first considered the toy datasets described in section 4.1 of AdaGAN [Tolstikhin et al., 2017], where we can explicitly compare various GANs with well-defined, likelihood-based performance metrics. The true data distribution is a mixture of 9 isotropic Gaussian components on $\mathcal{X} = \mathbb{R}^2$, with their centers uniformly distributed on a circle. We used the AdaGAN algorithm to sequentially generate 10 GANs, and compared various ensembles of these 10 networks: $\text{GAN}_1$ generated by the baseline GAN algorithm; $\text{Ada}_5$ and $\text{Ada}_{10}$, generated by AdaGAN with the first 5 or 10 GANs, respectively; $\text{EDGAN}_5$ and $\text{EDGAN}_{10}$, the ensembles of the first 5 or 10 GANs by EDGAN, respectively.

The EDGAN algorithm ran with squared loss and linear mappings. To measure the performance, we computed the likelihood of the generated data under the true distribution $L(S_\theta)$, and the likelihood of the true data under the generated distribution $L(S_r)$. We used kernel density estimation with cross-validated bandwidth to approximate the density of both $\mathbb{P}_\theta$ and $\mathbb{P}_r$, as in Tolstikhin et al. [2017]. We provide part of the ensembles here and present the full results in Appendix C. Table 2 compares the two likelihood-based metrics averaged over 10 repetitions, with standard deviation in parentheses.

Table 2: Likelihood-based metrics of various ensembles of 10 GANs.

| | $L(S_r)$ | $L(S_\theta)$ |
|---|---|---|
| $GAN_1$ | -12.39 ($\pm$ 2.12) | -796.05 ($\pm$ 12.48) |
| $Ada_{10}$ | -4.33 ($\pm$ 0.30) | -266.60 ($\pm$ 24.91) |
| $EDGAN_{10}$ | -3.99 ($\pm$ 0.20) | -148.97 ($\pm$ 14.13) |

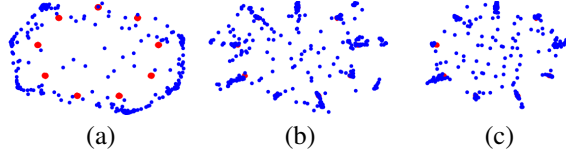

(a)      (b)      (c)

Figure 6: The true (red) and the generated (blue) distributions using (a) $GAN_1$; (b) $Ada_{10}$; (c) $EDGAN_{10}$.

Table 3: Each row uses a different embedding to calculate the discrepancy between the generated images and the CIFAR10 test set.

| | $GAN_1$ | $GAN_2$ | $GAN_3$ | $GAN_4$ | $GAN_5$ | Best GAN | Average | EDGAN |
|---|---|---|---|---|---|---|---|---|
| InceptionLogits | 285.09 | 259.61 | 259.64 | 271.21 | 272.23 | 259.61 | 259.12 | **255.3** |
| InceptionPool | 70.52 | 64.37 | 69.48 | 69.69 | 68.7 | 64.37 | 66.08 | **63.98** |
| MobileNet | 109.09 | 90.47 | 88.01 | 90.9 | 93.08 | 88.01 | 85.71 | **81.83** |
| PNASNet | 35.18 | 36.42 | 34.94 | 34.38 | 36.52 | 34.38 | 34.66 | **33.97** |
| NASNet | 54.61 | 52.66 | 59.01 | 61.79 | 64.97 | 52.66 | 55.66 | **52.46** |
| AmoebaNet | **97.71** | 110.83 | 108.61 | 105.31 | 110.5 | 97.71 | 104.91 | **97.71** |

We can see that for both metrics, ensembles of networks by EDGAN outperformed AdaGAN using the same number of base networks. Figure 6 shows the true distribution (in red) and the generated distribution (in blue). The single GAN model (Figure 6(a)) does not work well. As AdaGAN gradually mixes in more networks, the generated distribution is getting closer to the true distribution (Figure 6(b)). By explicitly learning the mixture weights using discrepancy, $EDGAN_{10}$ (Figure 6(c)) further improves over $Ada_{10}$, such that the span of the generated distribution is reduced, and the generated distribution now closely concentrates around the true one.

**CIFAR10** We used five pre-trained generators from Lucic et al. [2018] (all are publicly available on TF-Hub) as base learners in the ensemble. The models were trained with different hyperparameters and had different levels of performance. We then took 50k samples from each generator and the training split of CIFAR10, and embedded these images using a pre-trained classifier. We used several embeddings: InceptionV3's logits layer [Szegedy et al., 2016], InceptionV3's pooling layer [Szegedy et al., 2016], MobileNet [Sandler et al., 2018], PNASNet Liu et al. [2018a], NASNet [Zoph and Le, 2017], and AmoebaNet [Real et al., 2019]. All of these models are also available on TF-Hub. For each embedding, we trained an ensemble and evaluated its discrepancy on the test set of CIFAR10 and 10k independent samples from each generator. We report these results in Table 3. In all cases EDGAN performs as well or better than the best individual generator or a uniform average of the generators. This also shows that discrepancy generalizes well from the training to the testing data. Interestingly, depending on which embedding is used for the ensemble, drastically different mixture weights are optimal, which demonstrates the importance of the hypothesis class for discrepancy. We list the learned ensemble weights in Table 5 in Appendix C.

## 5 Conclusion

We advocated the use of discrepancy for defining GANs and proved a series of favorable properties for it, including continuity, under mild assumptions, the possibility of accurately estimating it from finite samples, and the generalization guarantees it benefits from. We also showed empirically that DGAN is competitive with other GANs, and that EDGAN, which we showed can be formulated as a convex optimization problem, outperforms existing GAN ensembles. For future work, one can use generative models with discrepancy in adaptation, as shown in Appendix D, where the goal is to learn a feature embedding for the target domain such that its distribution is close to the distribution of the embedded source domain. DGAN also has connections with standard Maximum Entropy models (Maxent) as discussed in Appendix E.

**Acknowledgments**

This work was partly supported by NSF CCF-1535987, NSF IIS-1618662, and a Google Research Award. We thank Judy Hoffman for helpful pointers to the literature.

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
