[Supplementary Material]

# A  GAN and WGAN

In this appendix section, we briefly introduce and discuss two instances of the distance metric $d$, which lead to two widely-used GANs: the original GAN [Goodfellow et al., 2014], and the WGAN [Arjovsky et al., 2017]. Note that in practice, often the value of $d(\mathbb{P}, \mathbb{Q})$ is not directly computable, and its variational form is used instead.

## A.1  GAN: Jensen-Shannon divergence

Goodfellow et al. [2014] introduced the first GAN framework using the *Jensen-Shannon* divergence:

$$d(\mathbb{P}_r, \mathbb{P}_\theta) := \mathrm{JS}(\mathbb{P}_r, \mathbb{P}_\theta) = \big( \mathrm{KL}(\mathbb{P}_r \parallel \mathbb{P}_m) + \mathrm{KL}(\mathbb{P}_\theta \parallel \mathbb{P}_m) \big)/2,$$

where $\mathbb{P}_m = (\mathbb{P}_r + \mathbb{P}_\theta)/2$. The Jensen-Shannon divergence admits the following equivalent form:

$$\mathrm{JS}(\mathbb{P}_r, \mathbb{P}_\theta) = \sup_{f \colon \mathcal{X} \to [0,1]} \frac{1}{2} \left\{ \mathop{\mathbb{E}}_{x \sim \mathbb{P}_r} \big[ \log f(x) \big] + \mathop{\mathbb{E}}_{x \sim \mathbb{P}_\theta} \big[ \log(1 - f(x)) \big] + \log 4 \right\}. \tag{4}$$

GANs were originally motivated by expression (4), and its equivalence to the Jensen-Shannon divergence was shown later on. Think of $f$ in equation (4) as a "discriminator" trying to tell apart real data from "fake" data generated by $g_\theta$ as follows: $f$ gives higher scores to samples which it thinks are real, and gives lower scores otherwise. Thus the maximization in (4) looks for the best discriminator $f$. On the other hand, the generator $g_\theta$ tries to fool the discriminator $f$, such that $f$ cannot tell the difference between real and fake samples. Thus minimizing $\mathrm{JS}(\mathbb{P}_r, \mathbb{P}_\theta)$ over $\theta$ looks for the best generator $g_\theta$. Dropping the constants in (4) and parametrizing the discriminator $f$ with a family of neural networks $\{f_w \colon \mathcal{X} \to [0,1], w \in W\}$, the original algorithm of training GAN Goodfellow et al. [2014] considers the following min-max optimization problem:

$$\min_{\theta \in \Theta} \max_{w \in W} \left\{ \mathop{\mathbb{E}}_{x \sim \mathbb{P}_r} \big[ \log f_w(x) \big] + \mathop{\mathbb{E}}_{x \sim \mathbb{P}_\theta} \big[ \log(1 - f_w(x)) \big] \right\}. \tag{5}$$

The generator and the discriminator $(g_\theta, f_w)$ are trained via stochastic gradient descent/ascent on objective (5) with respect to $\theta$ and $w$ iteratively, using the empirical distributions $\widehat{\mathbb{P}}_r$ and $\widehat{\mathbb{P}}_\theta$ induced by the batch of samples at each step.

**Remark 1.** *When minimizing $\theta$, one can drop the first term in (5) since it does not depend on $\theta$. Thus, for a fixed $w$, the minimizing step of $\theta$ is equivalent to $\min_\theta \mathbb{E}_{x \sim \mathbb{P}_\theta} \big[ \log(1 - f_w(x)) \big]$. Goodfellow et al. [2014] suggested using maximization instead of minimization to speed up training for $\theta$. That is, the training of GAN iterates between*

$$\max_{w \in W} \left\{ \mathop{\mathbb{E}}_{x \sim \mathbb{P}_r} \big[ \log f_w(x) \big] + \mathop{\mathbb{E}}_{x \sim \mathbb{P}_\theta} \big[ \log(1 - f_w(x)) \big] \right\},$$

$$\max_{\theta \in \Theta} \left\{ \mathop{\mathbb{E}}_{x \sim \mathbb{P}_\theta} \big[ \log f_w(x) \big] \right\}.$$

## A.2  WGAN: Wasserstein Distance

Instead of minimizing $\mathrm{JS}(\mathbb{P}_r, \mathbb{P}_\theta)$, Arjovsky et al. [2017] proposed to use the *Wasserstein* distance:

$$d(\mathbb{P}_r, \mathbb{P}_\theta) := W(\mathbb{P}_r, \mathbb{P}_\theta) = \inf_{\gamma \in \Pi(\mathbb{P}_r, \mathbb{P}_\theta)} \left\{ \mathop{\mathbb{E}}_{(x,y) \sim \gamma} \|x - y\| \right\},$$

where $\Pi(\mathbb{P}_r, \mathbb{P}_\theta) = \{\gamma(x,y) \colon \int_y \gamma(x,y)dy = \mathbb{P}_r(x), \int_x \gamma(x,y)dx = \mathbb{P}_\theta(y)\}$ denotes the set of joint distributions whose marginals are $\mathbb{P}_r$ and $\mathbb{P}_\theta$, respectively. By the Kantorovich-Rubinstein duality [Villani, 2008], Wasserstein distance can also be written as

$$W(\mathbb{P}_r, \mathbb{P}_\theta) = \sup_{\|f\|_L \leq 1} \left\{ \mathop{\mathbb{E}}_{x \sim \mathbb{P}_r} \big[ f(x) \big] - \mathop{\mathbb{E}}_{x \sim \mathbb{P}_\theta} \big[ f(x) \big] \right\}, \tag{6}$$

where the supremum is taken over all 1-Lipschitz functions with respect to the metric $\|\cdot\|$ that defines $W(\mathbb{P}_r, \mathbb{P}_\theta)$. Again, we can view $f$ as the discriminator, which aims to maximizes the difference between its expected values on the real data and that on the fake data. In practice, Arjovsky et al. [2017] set $f$ to be a neural networks whose parameters $w$ are limited within a compact set $W$, thus

$\{f_w \colon w \in W\}$ is a set of $K$-Lipschitz functions for some constant $K$. Thus, the WGAN (Wasserstein GAN) considered the following min-max optimization problem:

$$\min_{\theta \in \Theta} \max_{w \in W} \left\{ \mathop{\mathbb{E}}_{x \sim \mathbb{P}_r} \big[ f_w(x) \big] - \mathop{\mathbb{E}}_{x \sim \mathbb{P}_\theta} \big[ f_w(x) \big] \right\}. \tag{7}$$

The training procedure of WGANs is similar to that of GANs [Goodfellow et al., 2014], where one optimizes objective (7) using mini-batches with respect to $\theta$ and $w$ iteratively.

Arjovsky et al. [2017] showed that the JS divergence of GAN is potentially not continuous with respect to generator's parameter $\theta$. On the other hand, under mild conditions, $W(\mathbb{P}_r, \mathbb{P}_\theta)$ is continuous everywhere and differentiable almost everywhere with respect to $\theta$, making it easier to train WGAN.

When training WGAN with (7), one need to clip the weights $w$ to ensure that $w \in W$. More recently, Gulrajani et al. [2017] found that weight clipping can lead to undesired behavior, such as capacity underuse, and exploding or vanishing gradients. In view of this, they proposed to add a gradient penalty to the objective of WGAN, as an alternative to weight clipping:

$$\min_{\theta \in \Theta} \max_{w} \left\{ \mathop{\mathbb{E}}_{x \sim \mathbb{P}_r} \big[ f_w(x) \big] - \mathop{\mathbb{E}}_{x \sim \mathbb{P}_\theta} \big[ f_w(x) \big] + \lambda \mathop{\mathbb{E}}_{x \sim \mathbb{P}_l} \big[ \left( \| \nabla_x f_w(x) \|_2 - 1 \right)^2 \big] \right\},$$

where $\mathbb{P}_l$ indicates the uniform distribution along straight lines between pairs of points in $S_r$ and $S_\theta$. The construction of $\mathbb{P}_l$ is motivated by the optimality conditions.

WGAN is closely related to DGAN. The definitions of Wasserstein distance (6) and discrepancy (1) are syntactically the same, except that the former takes supremum over all 1-Lipschitz functions, while the latter takes supremum over $\ell_{\mathcal{H}} = \big\{ \ell\big( h(x), h'(x) \big) \colon h, h' \in \mathcal{H} \big\}$, a set that depends on the loss and hypothesis set. Thus, Wasserstein distance can be viewed as discrepancy without the hypothesis set and the loss function, which is one reason it cannot benefit from theoretical guarantees.

## B  Proofs

**Theorem 1.** *Assume the true labeling function $f \colon \mathcal{X} \to \mathcal{Y}$ is contained in the hypothesis set $\mathcal{H}$. Then, for any hypothesis $h \in \mathcal{H}$,*

$$\mathop{\mathbb{E}}_{x \sim \mathbb{P}_r} [\ell(h, f)] \leq \mathop{\mathbb{E}}_{x \sim \mathbb{P}_\theta} [\ell(h, f)] + disc_{\mathcal{H}, \ell}(\mathbb{P}_\theta, \mathbb{P}_r).$$

*Proof.* By the definition of $disc_{\mathcal{H}, \ell}$, for any $h \in \mathcal{H}$,

$$
\begin{aligned}
\mathop{\mathbb{E}}_{x \sim \mathbb{P}_r} [\ell(h, f)] &\leq \mathop{\mathbb{E}}_{x \sim \mathbb{P}_\theta} [\ell(h, f)] + \Big| \mathop{\mathbb{E}}_{x \sim \mathbb{P}_r} [\ell(h, f)] - \mathop{\mathbb{E}}_{x \sim \mathbb{P}_\theta} [\ell(h, f)] \Big| \\
&\leq \mathop{\mathbb{E}}_{x \sim \mathbb{P}_\theta} [\ell(h, f)] + \sup_{h, h' \in \mathcal{H}} \Big| \mathop{\mathbb{E}}_{x \sim \mathbb{P}_r} [\ell(h, h')] - \mathop{\mathbb{E}}_{x \sim \mathbb{P}_\theta} [\ell(h, h')] \Big| \qquad \text{(Since } f \in \mathcal{H}) \\
&= \mathop{\mathbb{E}}_{x \sim \mathbb{P}_\theta} [\ell(h, f)] + disc_{\mathcal{H}, \ell}(\mathbb{P}_\theta, \mathbb{P}_r).
\end{aligned}
$$

$\square$

**Proposition 6.** *Let $\mathcal{H}$ be a set of 1-Lipschitz functions. Then, $\ell_{\mathcal{H}} = \big\{ \big[ h(x) - h'(x) \big]^2 \colon h, h' \in \mathcal{H} \big\}$ is a set of Lipschitz functions on $\{x \colon \|x\| \leq 1\}$ with Lipschitz constant equals $4$.*

*Proof.* By definition,

$$
\begin{aligned}
|f(x) - f(x')| &= \Big| \big[ h(x) - h'(x) \big]^2 - \big[ h(x') - h'(x') \big]^2 \Big| \\
&\leq 2 \big| |h(x) - h'(x)| - |h(x') - h'(x')| \big| && (\ell_2 \text{ loss is 2-Lipschitz}) \\
&\leq 2 \big| h(x) - h'(x) - h(x') + h'(x') \big| && \text{(Triangle inequality)} \\
&\leq 2 \big| h(x) - h(x') \big| + 2 \big| h'(x) - h'(x') \big| && \text{(Triangle inequality)} \\
&\leq 4 \| x - x' \|. && (h \text{ and } h' \text{ are 1-Lipschitz})
\end{aligned}
$$

$\square$

**Theorem 2.** *Assume that $\mathcal{H} = \{h\colon \mathcal{X} \to \mathcal{Y}\}$ is a family of $\mu$-Lipschitz functions, and the loss function $\ell$ is continuous and symmetric in its arguments, and bounded by $M$. Furthermore, $\ell$ admits the triangle inequality, or it can be written as $\ell(y, y') = f(|y - y'|)$ for some Lipschitz function $f$. Assume that $g_\theta\colon \mathcal{Z} \to \mathcal{X}$ is continuous in $\theta$. Then, $disc_{\mathcal{H},\ell}(\mathbb{P}_r, \mathbb{P}_\theta)$ is continuous in $\theta$.*

*Proof.* We first consider the case where $\ell$ admits triangle inequality. We will show that $disc_{\mathcal{H},\ell}(\mathbb{P}_\theta, \mathbb{P}_{\theta'}) \to 0$ as $\theta \to \theta'$. By definition of $\mathbb{P}_\theta$ and $disc_{\mathcal{H},\ell}$,

$$
disc_{\mathcal{H},\ell}(\mathbb{P}_\theta, \mathbb{P}_{\theta'}) = \sup_{h,h' \in \mathcal{H}} \left| \mathbb{E}_{z \sim \mathbb{P}_z} \left[ \ell\Big(h\big(g_\theta(z)\big), h'\big(g_\theta(z)\big)\Big) - \ell\Big(h\big(g_{\theta'}(z)\big), h'\big(g_{\theta'}(z)\big)\Big) \right] \right|
$$

$$
\leq \sup_{h,h' \in \mathcal{H}} \mathbb{E}_{z \sim \mathbb{P}_z} \left| \ell\Big(h\big(g_\theta(z)\big), h'\big(g_\theta(z)\big)\Big) - \ell\Big(h\big(g_{\theta'}(z)\big), h'\big(g_{\theta'}(z)\big)\Big) \right|
$$

$$
\leq \sup_{h,h' \in \mathcal{H}} \mathbb{E}_{z \sim \mathbb{P}_z} \left[ \ell\Big(h\big(g_\theta(z)\big), h\big(g_{\theta'}(z)\big)\Big) + \ell\Big(h'\big(g_\theta(z)\big), h'\big(g_{\theta'}(z)\big)\Big) \right],
$$

where we used the triangle inequality and symmetry of $\ell$, such that $\forall a, b, c, d \in \mathcal{Y}$,

$$
|\ell(a, b) - \ell(c, d)| \leq \ell(a, c) + \ell(b, d).
$$

Thus,

$$
disc_{\mathcal{H},\ell}(\mathbb{P}_\theta, \mathbb{P}_{\theta'}) \leq \sup_{h \in \mathcal{H}} 2\, \mathbb{E}_{z \sim \mathbb{P}_z} \left[ \ell\Big(h\big(g_\theta(z)\big), h\big(g_{\theta'}(z)\big)\Big) \right] \leq 2\, \mathbb{E}_{z \sim \mathbb{P}_z} \left[ \sup_{h \in \mathcal{H}} \ell\Big(h\big(g_\theta(z)\big), h\big(g_{\theta'}(z)\big)\Big) \right].
$$

Since $\forall h \in \mathcal{H}$ is $L$-Lipschitz, for any $x_0 \in \mathcal{X}$,

$$
\lim_{x \to x_0} \ell\Big(h(x), h(x_0)\Big) = 0
$$

converges uniformly over $h \in \mathcal{H}$. Furthermore, $g_\theta$ is continuous in $\theta$, it follows that for any fixed $z \in \mathcal{Z}$,

$$
\lim_{\theta \to \theta'} \sup_{h \in \mathcal{H}} \ell\Big(h\big(g_\theta(z)\big), h\big(g_{\theta'}(z)\big)\Big) = 0,
$$

thus converges point-wise as functions of $z$. Since $\ell \leq M$ is bounded, by bounded convergence theorem, we have

$$
\lim_{\theta \to \theta'} disc_{\mathcal{H},\ell}(\mathbb{P}_\theta, \mathbb{P}_{\theta'}) \leq 2 \lim_{\theta \to \theta'} \mathbb{E}_{z \sim \mathbb{P}_z} \left[ \sup_{h \in \mathcal{H}} \ell\Big(h\big(g_\theta(z)\big), h\big(g_{\theta'}(z)\big)\Big) \right] = 0.
$$

Now we consider the case where $\ell(a, b) = f(|a - b|)$, and $f$ is a $q$-Lipschitz function: $|f(x) - f(x')| \leq q|x - x'|$. By definition,

$$
disc_{\mathcal{H},\ell}(\mathbb{P}_\theta, \mathbb{P}_{\theta'}) = \sup_{h,h' \in \mathcal{H}} \left| \mathbb{E}_{z \sim \mathbb{P}_z} \left[ \ell\Big(h\big(g_\theta(z)\big), h'\big(g_\theta(z)\big)\Big) - \ell\Big(h\big(g_{\theta'}(z)\big), h'\big(g_{\theta'}(z)\big)\Big) \right] \right|
$$

$$
\leq \sup_{h,h' \in \mathcal{H}} \mathbb{E}_{z \sim \mathbb{P}_z} \left| \ell\Big(h\big(g_\theta(z)\big), h'\big(g_\theta(z)\big)\Big) - \ell\Big(h\big(g_{\theta'}(z)\big), h'\big(g_{\theta'}(z)\big)\Big) \right|
$$

$$
= \sup_{h,h' \in \mathcal{H}} \mathbb{E}_{z \sim \mathbb{P}_z} \left| f\Big(\big|h\big(g_\theta(z)\big) - h'\big(g_\theta(z)\big)\big|\Big) - f\Big(\big|h\big(g_{\theta'}(z)\big) - h'\big(g_{\theta'}(z)\big)\big|\Big) \right|
$$

$$
\leq q \sup_{h,h' \in \mathcal{H}} \mathbb{E}_{z \sim \mathbb{P}_z} \left| \big|h\big(g_\theta(z)\big) - h'\big(g_\theta(z)\big)\big| - \big|h\big(g_{\theta'}(z)\big) - h'\big(g_{\theta'}(z)\big)\big| \right|
$$

$$
\leq q \sup_{h,h' \in \mathcal{H}} \mathbb{E}_{z \sim \mathbb{P}_z} \left| h\big(g_\theta(z)\big) - h\big(g_{\theta'}(z)\big) - h'\big(g_\theta(z)\big) + h'\big(g_{\theta'}(z)\big) \right|
$$

$$
\leq q \sup_{h,h' \in \mathcal{H}} \mathbb{E}_{z \sim \mathbb{P}_z} \left[ \big|h\big(g_\theta(z)\big) - h\big(g_{\theta'}(z)\big)\big| + \big|h'\big(g_\theta(z)\big) - h'\big(g_{\theta'}(z)\big)\big| \right]
$$

$$
= 2q \sup_{h \in \mathcal{H}} \mathbb{E}_{z \sim \mathbb{P}_z} \big|h\big(g_\theta(z)\big) - h\big(g_{\theta'}(z)\big)\big|
$$

$$
\leq 2q\, \mathbb{E}_{z \sim \mathbb{P}_z} \sup_{h \in \mathcal{H}} \big|h\big(g_\theta(z)\big) - h\big(g_{\theta'}(z)\big)\big|.
$$

Then, by the same argument above,

$$\lim_{\theta \to \theta'} \mathrm{disc}_{\mathcal{H},\ell}(\mathbb{P}_\theta, \mathbb{P}_{\theta'}) \leq 2q \lim_{\theta \to \theta'} \mathop{\mathbb{E}}_{z \sim \mathbb{P}_z} \sup_{h \in \mathcal{H}} \left| h\big(g_\theta(z)\big), h\big(g_{\theta'}(z)\big) \right| = 0.$$

Finally, by the triangle inequality of $\mathrm{disc}_{\mathcal{H},\ell}$, $\mathrm{disc}_{\mathcal{H},\ell}(\mathbb{P}_r, \mathbb{P}_\theta) - \mathrm{disc}_{\mathcal{H},\ell}(\mathbb{P}_r, \mathbb{P}_{\theta'}) \leq \mathrm{disc}_{\mathcal{H},\ell}(\mathbb{P}_\theta, \mathbb{P}_{\theta'})$, which completes the proof. ☐

**Theorem 3.** *Assume the loss is bounded, $\ell \leq M$. For any $\delta > 0$, with probability at least $1 - \delta$ over the drawn of $S_r$ and $S_\theta$,*

$$\left| disc_{\mathcal{H},\ell}(\mathbb{P}_r, \mathbb{P}_\theta) - disc_{\mathcal{H},\ell}(\widehat{\mathbb{P}}_r, \widehat{\mathbb{P}}_\theta) \right| \leq \widehat{\mathfrak{R}}_{S_r}(\ell_{\mathcal{H}}) + \widehat{\mathfrak{R}}_{S_\theta}(\ell_{\mathcal{H}}) + 3M\left( \sqrt{\tfrac{\log(4/\delta)}{2m}} + \sqrt{\tfrac{\log(4/\delta)}{2n}} \right).$$

*Furthermore, when the loss function $\ell(h, h')$ is a $q$-Lipschitz function of $h - h'$, we have*

$$\left| disc_{\mathcal{H},\ell}(\mathbb{P}_r, \mathbb{P}_\theta) - disc_{\mathcal{H},\ell}(\widehat{\mathbb{P}}_r, \widehat{\mathbb{P}}_\theta) \right| \leq 4q\left( \widehat{\mathfrak{R}}_{S_r}(\mathcal{H}) + \widehat{\mathfrak{R}}_{S_\theta}(\mathcal{H}) \right) + 3M\left( \sqrt{\tfrac{\log(4/\delta)}{2m}} + \sqrt{\tfrac{\log(4/\delta)}{2n}} \right).$$

*Proof.* By triangle inequality of $\mathrm{disc}_{\mathcal{H},\ell}(\cdot, \cdot)$,

$$|\mathrm{disc}_{\mathcal{H},\ell}(\mathbb{P}_r, \mathbb{P}_\theta) - \mathrm{disc}_{\mathcal{H},\ell}(\widehat{\mathbb{P}}_r, \widehat{\mathbb{P}}_\theta)| \leq \mathrm{disc}_{\mathcal{H},\ell}(\mathbb{P}_r, \widehat{\mathbb{P}}_r) + \mathrm{disc}_{\mathcal{H},\ell}(\mathbb{P}_\theta, \widehat{\mathbb{P}}_\theta).$$

We first apply concentration inequality to the scaled loss $\ell_{\mathcal{H}}/M$:

$$\mathop{\mathbb{E}}_{\mathbb{P}_r} \ell(h, h')/M \leq \mathop{\mathbb{E}}_{\widehat{\mathbb{P}}_r} \ell(h, h')/M + \widehat{\mathfrak{R}}_{S_r}(\ell_{\mathcal{H}}/M) + 3\sqrt{\frac{\log(4/\delta)}{2m}},$$

$$\mathop{\mathbb{E}}_{\mathbb{P}_\theta} \ell(h, h')/M \leq \mathop{\mathbb{E}}_{\widehat{\mathbb{P}}_\theta} \ell(h, h')/M + \widehat{\mathfrak{R}}_{S_\theta}(\ell_{\mathcal{H}}/M).$$

For the empirical Radmacher complexity, we have $\widehat{\mathfrak{R}}_{c\mathcal{H}} = c\widehat{\mathfrak{R}}_{\mathcal{H}}$. Thus, we have

$$|\mathrm{disc}_{\mathcal{H},\ell}(\mathbb{P}_r, \mathbb{P}_\theta) - \mathrm{disc}_{\mathcal{H},\ell}(\widehat{\mathbb{P}}_r, \widehat{\mathbb{P}}_\theta)|$$

$$\leq \mathrm{disc}_{\mathcal{H},\ell}(\mathbb{P}_r, \widehat{\mathbb{P}}_r) + \mathrm{disc}_{\mathcal{H},\ell}(\mathbb{P}_\theta, \widehat{\mathbb{P}}_\theta) \leq \widehat{\mathfrak{R}}_{S_r}(\ell_{\mathcal{H}}) + \widehat{\mathfrak{R}}_{S_\theta}(\ell_{\mathcal{H}}) + 3M\left( \sqrt{\frac{\log(4/\delta)}{2m}} + \sqrt{\frac{\log(4/\delta)}{2n}} \right).$$

When the loss function $\ell(h, h')$ is a $q$-Lipschitz function of the difference of its two arguments, i.e. $\ell(a, b) = f(a - b)$, and $f(\cdot)$ is a $q$-Lipschitz function, the mapping of $\mathcal{H} \ominus \mathcal{H} \to \ell_{\mathcal{H}}$ is $q$-Lipschitz, where $\mathcal{H} \ominus \mathcal{H}$ is defined as $\mathcal{H} \ominus \mathcal{H} = \{h - h' \colon h, h' \in \mathcal{H}\}$. By Talagrand's contraction lemma, $\widehat{\mathfrak{R}}_{\ell_{\mathcal{H}}} \leq 2q\mathfrak{R}_{\mathcal{H} \ominus \mathcal{H}}$. Finally, by definition we have $\widehat{\mathfrak{R}}_{\mathcal{H} \ominus \mathcal{H}} \leq 2\widehat{\mathfrak{R}}_{\mathcal{H}}$. Putting everything together, when the loss function $\ell(h, h')$ is a $q$-Lipschitz function of $h - h'$,

$$|\mathrm{disc}_{\mathcal{H},\ell}(\mathbb{P}_r, \mathbb{P}_\theta) - \mathrm{disc}_{\mathcal{H},\ell}(\widehat{\mathbb{P}}_r, \widehat{\mathbb{P}}_\theta)| \leq 4q\left( \widehat{\mathfrak{R}}_{S_r}(\mathcal{H}) + \widehat{\mathfrak{R}}_{S_\theta}(\mathcal{H}) \right) + 3M\left( \sqrt{\frac{\log(4/\delta)}{2m}} + \sqrt{\frac{\log(4/\delta)}{2n}} \right).$$

☐

**Theorem 5.** *For any $\delta > 0$, with probability at least $1 - \delta$ over the draw of samples,*

$$\left| disc_{\mathcal{H},\ell}(\mathbb{P}_{\widehat{\alpha}}, \mathbb{P}_r) - disc_{\mathcal{H},\ell}(\mathbb{P}_{\alpha^*}, \mathbb{P}_r) \right| \leq 2\left( \widehat{\mathfrak{R}}_S(\ell_{\mathcal{H}}) + 3M\sqrt{\log[4(p+1)/\delta]/2n} \right),$$

*where $\widehat{\mathfrak{R}}_S(\ell_{\mathcal{H}}) = \max\left\{ \widehat{\mathfrak{R}}_{S_1}(\ell_{\mathcal{H}}), \ldots, \widehat{\mathfrak{R}}_{S_p}(\ell_{\mathcal{H}}), \widehat{\mathfrak{R}}_{S_r}(\ell_{\mathcal{H}}) \right\}$. Furthermore, when the loss function $\ell(h, h')$ is a $q$-Lipschitz function of $h - h'$, the following holds with probability $1 - \delta$:*

$$\left| disc_{\mathcal{H},\ell}(\mathbb{P}_{\widehat{\alpha}}, \mathbb{P}_r) - disc_{\mathcal{H},\ell}(\mathbb{P}_{\alpha^*}, \mathbb{P}_r) \right| \leq 2\left( 4q\widehat{\mathfrak{R}}_S(\mathcal{H}) + 3M\sqrt{\log[4(p+1)/\delta]/2n} \right),$$

*where $\widehat{\mathfrak{R}}_S(\mathcal{H}) = \max\left\{ \widehat{\mathfrak{R}}_{S_1}(\mathcal{H}), \ldots, \widehat{\mathfrak{R}}_{S_p}(\mathcal{H}), \widehat{\mathfrak{R}}_{S_r}(\mathcal{H}) \right\}$.*

*Proof.* We first extend Theorem 3 to the case of GAN ensembles:

$$|\text{disc}_{\mathcal{H},\ell}(\mathbb{P}_{\boldsymbol{\alpha}}, \mathbb{P}_r) - \text{disc}_{\mathcal{H},\ell}(\widehat{\mathbb{P}}_{\boldsymbol{\alpha}}, \widehat{\mathbb{P}}_r)| \leq \text{disc}_{\mathcal{H},\ell}(\mathbb{P}_{\boldsymbol{\alpha}}, \widehat{\mathbb{P}}_{\boldsymbol{\alpha}}) + \text{disc}_{\mathcal{H},\ell}(\mathbb{P}_r, \widehat{\mathbb{P}}_r).$$

For the first term,

$$\text{disc}_{\mathcal{H},\ell}(\mathbb{P}_{\boldsymbol{\alpha}}, \widehat{\mathbb{P}}_{\boldsymbol{\alpha}})$$

$$\leq \sup_{h,h'\in\mathcal{H}} \Big| \sum_{k=1}^{p} \alpha_k \Big\{ \mathbb{E}_{x\sim\mathbb{P}_k} \big[\ell\big(h(x),h'(x)\big)\big] - \mathbb{E}_{x\sim\widehat{\mathbb{P}}_k} \big[\ell\big(h(x),h'(x)\big)\big] \Big\} \Big|$$

$$\leq \sup_{h,h'\in\mathcal{H}} \sum_{k=1}^{p} \alpha_k \Big| \mathbb{E}_{x\sim\mathbb{P}_k} \big[\ell\big(h(x),h'(x)\big)\big] - \mathbb{E}_{x\sim\widehat{\mathbb{P}}_k} \big[\ell\big(h(x),h'(x)\big)\big] \Big|$$

$$\leq \sum_{k=1}^{p} \alpha_k \sup_{h,h'\in\mathcal{H}} \Big| \mathbb{E}_{x\sim\mathbb{P}_k} \big[\ell\big(h(x),h'(x)\big)\big] - \mathbb{E}_{x\sim\widehat{\mathbb{P}}_k} \big[\ell\big(h(x),h'(x)\big)\big] \Big|$$

$$= \sum_{k=1}^{p} \alpha_k \, \text{disc}_{\mathcal{H},\ell}(\mathbb{P}_k, \widehat{\mathbb{P}}_k).$$

By concentration argument, with probability at least $1-\delta$ over the drawn of samples,

$$\text{disc}_{\mathcal{H},\ell}(\mathbb{P}_r, \widehat{\mathbb{P}}_r) \leq \widehat{\mathfrak{R}}_{S_r}(\ell_{\mathcal{H}}) + 3M\sqrt{\frac{\log(4(p+1)/\delta)}{2n}},$$

$$\text{disc}_{\mathcal{H},\ell}(\mathbb{P}_k, \widehat{\mathbb{P}}_k) \leq \widehat{\mathfrak{R}}_{S_k}(\ell_{\mathcal{H}}) + 3M\sqrt{\frac{\log(4(p+1)/\delta)}{2n}}.$$

Putting everything together, with probability at least $1-\delta$, for any $\alpha \in \Delta$,

$$|\text{disc}_{\mathcal{H},\ell}(\mathbb{P}_{\boldsymbol{\alpha}}, \mathbb{P}_r) - \text{disc}_{\mathcal{H},\ell}(\widehat{\mathbb{P}}_{\boldsymbol{\alpha}}, \widehat{\mathbb{P}}_r)|$$

$$\leq \sum_{k=1}^{n} \alpha_k \, \text{disc}_{\mathcal{H},\ell}(\mathbb{P}_k, \widehat{\mathbb{P}}_k) + \text{disc}_{\mathcal{H},\ell}(\mathbb{P}_r, \widehat{\mathbb{P}}_r)$$

$$\leq \sum_{k=1}^{n} \alpha_k \left[ \widehat{\mathfrak{R}}_{S_k}(\ell_{\mathcal{H}}) + 3M\sqrt{\frac{\log(4(p+1)/\delta)}{2n}} \right] + \widehat{\mathfrak{R}}_{S_r}(\ell_{\mathcal{H}}) + 3M\sqrt{\frac{\log(4(p+1)/\delta)}{2n}}$$

$$\leq \widehat{\mathfrak{R}}_S + 3M\sqrt{\frac{\log(4(p+1)/\delta)}{2n}}. \tag{8}$$

By definition,

$$\text{disc}_{\mathcal{H},\ell}(\mathbb{P}_{\boldsymbol{\alpha}^*}, \mathbb{P}_r) - \text{disc}_{\mathcal{H},\ell}(\mathbb{P}_{\widehat{\boldsymbol{\alpha}}}, \mathbb{P}_r)$$

$$\leq \text{disc}_{\mathcal{H},\ell}(\mathbb{P}_{\boldsymbol{\alpha}^*}, \mathbb{P}_r) - \text{disc}_{\mathcal{H},\ell}(\widehat{\mathbb{P}}_{\widehat{\boldsymbol{\alpha}}}, \widehat{\mathbb{P}}_r) + |\text{disc}_{\mathcal{H},\ell}(\widehat{\mathbb{P}}_{\widehat{\boldsymbol{\alpha}}}, \widehat{\mathbb{P}}_r) - \text{disc}_{\mathcal{H},\ell}(\mathbb{P}_{\widehat{\boldsymbol{\alpha}}}, \mathbb{P}_r)|$$

$$\leq \text{disc}_{\mathcal{H},\ell}(\mathbb{P}_{\widehat{\boldsymbol{\alpha}}}, \mathbb{P}_r) - \text{disc}_{\mathcal{H},\ell}(\widehat{\mathbb{P}}_{\widehat{\boldsymbol{\alpha}}}, \widehat{\mathbb{P}}_r) + |\text{disc}_{\mathcal{H},\ell}(\widehat{\mathbb{P}}_{\widehat{\boldsymbol{\alpha}}}, \widehat{\mathbb{P}}_r) - \text{disc}_{\mathcal{H},\ell}(\mathbb{P}_{\widehat{\boldsymbol{\alpha}}}, \mathbb{P}_r)|$$

$$\leq 2|\text{disc}_{\mathcal{H},\ell}(\widehat{\mathbb{P}}_{\widehat{\boldsymbol{\alpha}}}, \widehat{\mathbb{P}}_r) - \text{disc}_{\mathcal{H},\ell}(\mathbb{P}_{\widehat{\boldsymbol{\alpha}}}, \mathbb{P}_r)|$$

Similarly,

$$\text{disc}_{\mathcal{H},\ell}(\mathbb{P}_{\widehat{\boldsymbol{\alpha}}}, \mathbb{P}_r) - \text{disc}_{\mathcal{H},\ell}(\mathbb{P}_{\boldsymbol{\alpha}^*}, \mathbb{P}_r)$$

$$\leq \text{disc}_{\mathcal{H},\ell}(\mathbb{P}_{\widehat{\boldsymbol{\alpha}}}, \mathbb{P}_r) - \text{disc}_{\mathcal{H},\ell}(\widehat{\mathbb{P}}_{\boldsymbol{\alpha}^*}, \widehat{\mathbb{P}}_r) + |\text{disc}_{\mathcal{H},\ell}(\widehat{\mathbb{P}}_{\boldsymbol{\alpha}^*}, \widehat{\mathbb{P}}_r) - \text{disc}_{\mathcal{H},\ell}(\mathbb{P}_{\boldsymbol{\alpha}^*}, \mathbb{P}_r)|$$

$$\leq \text{disc}_{\mathcal{H},\ell}(\mathbb{P}_{\widehat{\boldsymbol{\alpha}}}, \mathbb{P}_r) - \text{disc}_{\mathcal{H},\ell}(\widehat{\mathbb{P}}_{\widehat{\boldsymbol{\alpha}}}, \widehat{\mathbb{P}}_r) + |\text{disc}_{\mathcal{H},\ell}(\widehat{\mathbb{P}}_{\boldsymbol{\alpha}^*}, \widehat{\mathbb{P}}_r) - \text{disc}_{\mathcal{H},\ell}(\mathbb{P}_{\boldsymbol{\alpha}^*}, \mathbb{P}_r)|$$

$$\leq |\text{disc}_{\mathcal{H},\ell}(\mathbb{P}_{\widehat{\boldsymbol{\alpha}}}, \mathbb{P}_r) - \text{disc}_{\mathcal{H},\ell}(\widehat{\mathbb{P}}_{\widehat{\boldsymbol{\alpha}}}, \widehat{\mathbb{P}}_r)| + |\text{disc}_{\mathcal{H},\ell}(\widehat{\mathbb{P}}_{\boldsymbol{\alpha}^*}, \widehat{\mathbb{P}}_r) - \text{disc}_{\mathcal{H},\ell}(\mathbb{P}_{\boldsymbol{\alpha}^*}, \mathbb{P}_r)|.$$

Thus, apply inequality (8) to $\widehat{\boldsymbol{\alpha}}$ and $\boldsymbol{\alpha}^*$, we have

$$|\text{disc}_{\mathcal{H},\ell}(\mathbb{P}_{\widehat{\boldsymbol{\alpha}}}, \mathbb{P}_r) - \text{disc}_{\mathcal{H},\ell}(\mathbb{P}_{\boldsymbol{\alpha}^*}, \mathbb{P}_r)| \leq 2\Big( \widehat{\mathfrak{R}}_S + 3M\sqrt{\frac{\log(4(p+1)/\delta)}{2n}} \Big).$$

$\square$

**Proposition 4.** *When $\ell$ is the squared loss and $\mathcal{H}$ the family of linear functions with norm bounded by 1, $disc_{\mathcal{H},\ell}(\widehat{\mathbb{P}}_r, \widehat{\mathbb{P}}_\theta) = 2 \left\| \frac{1}{n} X_\theta^T X_\theta - \frac{1}{m} X_r^T X_r \right\|_2$, where $\|\cdot\|_2$ denotes the spectral norm.*

*Proof.*

$$
\begin{aligned}
disc_{\mathcal{H},\ell}(\widehat{\mathbb{P}}_r, \widehat{\mathbb{P}}_\theta) &= \sup_{\substack{\|w\|_2 \leq 1 \\ \|w'\|_2 \leq 1}} \left| \mathop{\mathbb{E}}_{x \sim \widehat{\mathbb{P}}_r} \left[ \ell_2\left(w^T x, w'^T x\right) \right] - \mathop{\mathbb{E}}_{x \sim \widehat{\mathbb{P}}_\theta} \left[ \ell_2\left(w^T x, w'^T x\right) \right] \right| \\
&= \sup_{\substack{\|w\|_2 \leq 1 \\ \|w'\|_2 \leq 1}} \left| \frac{1}{n}(X_\theta w - X_\theta w')^T (X_\theta w - X_\theta w') - \frac{1}{m}(X_r w - X_r w')^T (X_r w - X_r w') \right| \\
&= \sup_{\|u\|_2 \leq 2} \left| \frac{1}{n} u^T X_\theta^T X_\theta u - \frac{1}{m} u^T X_r^T X_r u \right| \qquad\qquad \text{(Let } u = w - w') \\
&= 2 \sup_{\|u\|_2 \leq 1} \left| u^T \left( \frac{1}{n} X_\theta^T X_\theta - \frac{1}{m} X_r^T X_r \right) u \right| \\
&= 2 \left\| \frac{1}{n} X_\theta^T X_\theta - \frac{1}{m} X_r^T X_r \right\|_2 .
\end{aligned}
$$

$\square$

## C  More Experiments

### C.1  EDGAN: Toy datasets

In this section, we provide more results on mixing the 10 GANs generated by AdaGAN. Recall that we are comparing the following methods:

- The baseline GAN algorithm, namely $\text{GAN}_1$.
- The AdaGAN algorithm, ensembles of the first 5 GANs, namely $\text{Ada}_5$.
- The AdaGAN algorithm, ensembles of the first 10 GANs, namely $\text{Ada}_{10}$.
- The EDGAN algorithm, ensembles of the first 5 GANs, namely $\text{EDGAN}_5$.
- The EDGAN algorithm, ensembles of the first 10 GANs, namely $\text{EDGAN}_{10}$.

We considered two ways of computing average sample log-likelihood and used them as performance metrics: the likelihood of the generated data under the true distribution $L(S_\theta)$, and the likelihood of the true data under the generated distribution $L(S_r)$. To be more concrete,

$$
L(S_\theta) = L_{\mathbb{P}_r}(S_\theta) = \frac{1}{N} \sum_{x_i \in S_\theta} \log\left(\mathbb{P}_r(x_i)\right), \quad L(S_r) = L_{\mathbb{P}_\theta}(S_r) = \frac{1}{N} \sum_{x_i \in S_r} \log\left(\mathbb{P}_\theta(x_i)\right).
$$

We used kernel density estimation with cross-validated bandwidth to approximate the density of both $\mathbb{P}_\theta$ and $\mathbb{P}_r$.

Figure 7 displayed the true distribution (in red) and the generated distribution under various ensembles of GANs. $\text{EDGAN}_5$ and $\text{EDGAN}_{10}$ improve the generated distribution over $\text{Ada}_5$ and $\text{Ada}_{10}$, respectively.

Table 4 showed the average log-likelihoods over 10 repetitions, with standard deviations in parentheses, where a higher log-likelihood indicates better performance. We can see that for both metrics, networks ensembles $\text{EDGAN}_5$ and $\text{EDGAN}_{10}$ by EDGAN outperformed AdaGAN with the same number of base networks.

### C.2  EDGAN: CIFAR10

In this section, we provide the mixture weights of each ensemble when learning EDGAN on CIFAR10 generators, as described in Section 4.2. The data is provided in Table 5. Only in one instance a significant amount of the weight is allocated to one model.

Figure 7: The true (red) and the generated (blue) distributions, using various ensembles of 10 GANs.

Table 4: Likelihood-based metrics of various ensembles of 10 GANs.

|  | $L(S_r)$ | $L(S_\theta)$ |
|---|---|---|
| GAN$_1$ | -12.39 ($\pm$ 2.12) | -796.05 ($\pm$ 12.48) |
| Ada$_5$ | -5.02 ($\pm$ 0.11) | -296.45 ($\pm$ 15.24) |
| Ada$_{10}$ | -4.33 ($\pm$ 0.30) | -266.60 ($\pm$ 24.91) |
| EDGAN$_5$ | -4.85 ($\pm$ 0.16) | -172.52 ($\pm$ 17.56) |
| EDGAN$_{10}$ | -3.99 ($\pm$ 0.20) | -148.97 ($\pm$ 14.13) |

Table 5: The mixture weights of each ensemble.

|  | GAN$_1$ | GAN$_2$ | GAN$_3$ | GAN$_4$ | GAN$_5$ |
|---|---|---|---|---|---|
| InceptionLogits | 0.0007 | 0.4722 | 0.5252 | 0.0009 | 0.0009 |
| InceptionPool | 0.0042 | 0.7504 | 0.0139 | 0.0102 | 0.2213 |
| MobileNet | 0.0008 | 0.3718 | 0.3654 | 0.2416 | 0.0205 |
| PNasNet | 0.3325 | 0.0087 | 0.1400 | 0.5142 | 0.0044 |
| NasNet | 0.2527 | 0.7431 | 0.0021 | 0.0012 | 0.0009 |
| AmoebaNet | 0.9955 | 0.0005 | 0.0008 | 0.0026 | 0.0006 |

### C.3 Addition DGAN experimental details

All experiments for DGAN used Adam. On MNIST, we trained for 200 epochs at batch size 32 with learning rates of $3 \times 10^{-5}$ for the generator and $1 \times 10^{-5}$ for discriminator. For CIFAR10 and CIFAR 100, we trained for 256 epochs at batch size 32 with learning rates of $3 \times 10^{-5}$ for the generator and $1 \times 10^{-5}$ for discriminator. For CelebA, larger batch sizes and learning rates were necessary: batch size 256 and learning rates of $2 \times 10^{-4}$ for the generator and $5 \times 10^{-4}$ for discriminator.

## D  Domain Adaptation

We first introduce additional notation for the domain adaptation task. Let $\mathcal{D}_s$ and $\mathcal{D}_t$ denote the source and target distribution over $\mathcal{X} \times \mathcal{Y}$, and let $\widehat{\mathcal{D}}_s$ and $\widehat{\mathcal{D}}_t$ denote the empirical distribution induced by samples drawn according to $\mathcal{D}_s$ and $\mathcal{D}_t$, respectively. For any distribution $\mathcal{D}$, denote by $\mathcal{D}^{\mathcal{X}}$ its marginal distribution on the input space $\mathcal{X}$. Finally, for any marginal distribution $\mathcal{D}^{\mathcal{X}}$ and

Table 6: DGAN architectures based on Miyato et al. [2018]. Let $b$ be the batch size, $(h, w, c)$ be the shape of an input image, $f$ be the width of the discriminator's output embedding, and $g = 4$ for CIFAR and $g = 16$ for CelebA. The italicized layers in the discriminators are skipped for CIFAR (resulting in a shallower model), but are included for CelebA.

| **Discriminator** |
| :---: |
| Images $x \in \mathbb{R}^{b \times h \times w \times c}$ |
| $3 \times 3$, stride 1, conv. 64, BN, ReLU <br> $4 \times 4$, stride 2, conv. 64, BN, ReLU |
| $3 \times 3$, stride 1, conv. 128, BN, ReLU <br> $4 \times 4$, stride 2, conv. 128, BN, ReLU |
| *$3 \times 3$, stride 1, conv. 256, BN, ReLU* <br> *$4 \times 4$, stride 2, conv. 256, BN, ReLU* |
| $3 \times 3$, stride 1, conv. 512, BN, ReLU |
| dense$\rightarrow \mathbb{R}^{b \times f}$ |

| **Generator** |
| :---: |
| noise $z \in \mathbb{R}^{b \times 128}$ |
| dense$\rightarrow g^2 \times 512$ |
| $4 \times 4$, stride 2, deconv. 256, BN, ReLU |
| $4 \times 4$, stride 2, deconv. 128, BN, ReLU |
| $4 \times 4$, stride 2, deconv. 64, BN, ReLU |
| $3 \times 3$, stride 1, conv. 3, Tanh |

a feature mapping $M \colon \mathcal{X} \to \mathcal{Z}$ that maps input space $\mathcal{X}$ to some feature space $\mathcal{Z}$, we denote by $M(\mathcal{D}^{\mathcal{X}})$ the distribution of $M(x)$ where $x \sim \mathcal{D}^{\mathcal{X}}$.

### D.1 Adversarial Discriminative Domain Adaptation (ADDA)

Tzeng et al. [2017] considered a domain adaptation framework, Adversarial Discriminative Domain Adaptation (ADDA), which has a very similar motivation to GANs. Given a pre-trained source domain feature mapping $M_s$, ADDA simultaneously optimizes a target domain feature mapping $M_t$ and an adversarial discriminator, such that the best discriminator cannot tell apart the mapped features from source and target domain. At test time, ADDA applies the classifier trained on source feature mapping and labels to the learned target feature mapping, to predict target label. Take the multi-class classification task for example, the ADDA consists of three stages:

1. Pre-training. Given labeled samples $(X_s, Y_s)$ from source domain, learn a source feature mapping $M_s$ and a classifier $C$ under cross-entropy loss:

$$\min_{M_s, C} \left\{ - \mathop{\mathbb{E}}_{(x_s, y_s) \sim (X_s, Y_s)} \sum_{k=1}^{K} 1_{k=y_s} \log C(M_s(x_s)) \right\},$$

where $K$ is the number of label classes.

2. Adversarial adaptation. Given pre-trained source feature mapping $M_s$ and unlabeled samples $X_t$ from target domain, jointly learn a discriminator $D$ and a target feature mapping $M_t$:

$$\min_{D} \left\{ - \mathop{\mathbb{E}}_{x_s \sim X_s} \big[ \log D(M_s(x_s)) \big] - \mathop{\mathbb{E}}_{x_t \sim X_t} \big[ \log(1 - D(M_t(x_t))) \big] \right\}, \qquad \text{(Learn } D)$$

$$\min_{M_t} \left\{ - \mathop{\mathbb{E}}_{x_t \sim X_t} \big[ \log D(M_t(x_t)) \big] \right\}. \qquad \text{(Learn } M_t)$$

3. Testing. Predict label for target data based on $C(M_t(x_t))$.

Note that the second stage (adversarial adaptation) is very similar to the GAN framework, where the discriminator has the same functionality, and the generator is now mapping from the target data, instead of from a random latent variable, to a desired feature space.

The key idea of ADDA is very similar to GAN: the adversarial training step is in fact minimizing the Jensen-Shannon divergence between the mapped source distribution and the mapped target distribution:

$$\min_{M_t} \quad \text{JS}\Big( M_s(\mathcal{D}_s^{\mathcal{X}}), M_t(\mathcal{D}_t^{\mathcal{X}}) \Big).$$

Since discrepancy is originally designed for domain adaptation, it is natural to use discrepancy as the distance metric, instead of the Jensen-Shannon divergence, in this ADDA framework. We give more details below.

## D.2 DGAN for Domain Adaptation

The procedure of ADDA with discrepancy is very similar to the original ADDA, which is described below.

1. Pre-training. Given labeled samples from source domain, or equivalently an empirical distribution $\widehat{\mathcal{D}}_s$, learn a source feature mapping $M_s$ and a classifier $\widehat{h}_s \in \mathcal{H}$:

$$M_s, \widehat{h}_s = \operatorname*{argmin}_{M,h} \left\{ \mathop{\mathbb{E}}_{(x,y)\sim\widehat{\mathcal{D}}_s} \left[ \ell\Big(h(M(x)), y\Big) \right] \right\}$$

2. Adversarial adaptation. Given pre-trained source feature mapping $M_s$ and unlabeled samples from target domain (or equivalently, $\widehat{\mathcal{D}}_t^{\mathcal{X}}$), learn a target feature mapping $M_t$, such that the distribution of $M_s(\widehat{\mathcal{D}}_s^{\mathcal{X}})$ and $M_t(\widehat{\mathcal{D}}_t^{\mathcal{X}})$ are small under discrepancy:

$$M_t = \operatorname*{argmin}_{M} \left\{ \operatorname{disc}_{\mathcal{H},\ell}\Big( M_s(\widehat{\mathcal{D}}_s^{\mathcal{X}}), M(\widehat{\mathcal{D}}_t^{\mathcal{X}}) \Big) \right\}$$
$$= \operatorname*{argmin}_{M} \left\{ \sup_{h,h'\in\mathcal{H}} \Big| \mathop{\mathbb{E}}_{z\sim M_s(\widehat{\mathcal{D}}_s^{\mathcal{X}})} \Big[ \ell\Big(h(z), h'(z)\Big) \Big] - \mathop{\mathbb{E}}_{z\sim M(\widehat{\mathcal{D}}_t^{\mathcal{X}})} \Big[ \ell\Big(h(z), h'(z)\Big) \Big] \Big| \right\}. \tag{9}$$

3. Testing. Predict label for target data using $\widehat{h}_s(M_t(\cdot))$.

Suppose the target mapping $M_t$ is parameterized by and continuous in $\theta$. Then, under the same assumptions of Theorem 2, the objective function in (9) is continuous in $\theta$.

To analyze the adaptation performance, for the fixed mapping $M_s$ and $M_t$, we define the risk minimizers $h_s^*$ and $h_t^*$:

$$h_s^* = \operatorname*{argmin}_{h\in\mathcal{H}} \mathop{\mathbb{E}}_{(x,y)\sim\mathcal{D}_s} \left[ \ell\Big(h(M_s(x)), y\Big) \right], \quad h_t^* = \operatorname*{argmin}_{h\in\mathcal{H}} \mathop{\mathbb{E}}_{(x,y)\sim\mathcal{D}_t} \left[ \ell\Big(h(M_t(x)), y\Big) \right].$$

We have the following learning guarantees for ADDA with discrepancy.

**Theorem 7.** *Assume the loss function $\ell(\cdot, \cdot)$ is symmetric and obeys triangle inequality. Then,*

$$\mathop{\mathbb{E}}_{(x,y)\sim\mathcal{D}_t} \left[ \ell\Big(\widehat{h}_s(M_t(x)), y\Big) \right]$$
$$\leq \mathop{\mathbb{E}}_{x\sim\mathcal{D}_s^{\mathcal{X}}} \left[ \ell\Big(\widehat{h}_s(M_s(x)), h_s^*(M_s(x))\Big) \right] + \operatorname{disc}_{\mathcal{H},\ell}\Big( M_t(\mathcal{D}_t^{\mathcal{X}}), M_s(\mathcal{D}_s^{\mathcal{X}}) \Big)$$
$$+ \mathop{\mathbb{E}}_{x\sim\mathcal{D}_t^{\mathcal{X}}} \left[ \ell\Big(h_s^*(M_t(x)), h_t^*(M_t(x))\Big) \right] + \mathop{\mathbb{E}}_{(x,y)\sim\mathcal{D}_t} \left[ \ell\Big(h_t^*(M_t(x)), y\Big) \right].$$

*Proof.* By triangle inequality of $\ell$, we have

$$\mathop{\mathbb{E}}_{(x,y)\sim\mathcal{D}_t} \left[ \ell\Big(\widehat{h}_s(M_t(x)), y\Big) \right]$$
$$\leq \mathop{\mathbb{E}}_{(x,y)\sim\mathcal{D}_t} \left[ \ell\Big(\widehat{h}_s(M_t(x)), h_s^*(M_t(x))\Big) \right]$$
$$+ \mathop{\mathbb{E}}_{(x,y)\sim\mathcal{D}_t} \left[ \ell\Big(h_s^*(M_t(x)), h_t^*(M_t(x))\Big) \right] + \mathop{\mathbb{E}}_{(x,y)\sim\mathcal{D}_t} \left[ \ell\Big(h_t^*(M_t(x)), y\Big) \right]$$
$$\leq \mathop{\mathbb{E}}_{x\sim\mathcal{D}_s^{\mathcal{X}}} \left[ \ell\Big(\widehat{h}_s(M_s(x)), h_s^*(M_s(x))\Big) \right] + \operatorname{disc}_{\mathcal{H},\ell}\Big( M_t(\mathcal{D}_t^{\mathcal{X}}), M_s(\mathcal{D}_s^{\mathcal{X}}) \Big)$$
$$+ \mathop{\mathbb{E}}_{x\sim\mathcal{D}_t^{\mathcal{X}}} \left[ \ell\Big(h_s^*(M_t(x)), h_t^*(M_t(x))\Big) \right] + \mathop{\mathbb{E}}_{(x,y)\sim\mathcal{D}_t} \left[ \ell\Big(h_t^*(M_t(x)), y\Big) \right]$$

$\square$

Let us examine each item in Theorem 7:

- The first term is the estimation error of $\widehat{h}_s$, which should be small when a large set of source data is available.

- The second term is the true discrepancy between $M_t(\mathcal{D}_t^{\mathcal{X}})$ and $M_s(\mathcal{D}_s^{\mathcal{X}})$. According to Theorem 3, it can be accuracy estimated by its empirical counterparts, which is minimized during the training step (Equation (9)).

- The third term depends on how different $h_s^*$ and $h_t^*$ are, and it is essentially determined by how difficult the adaption problem is.

- The last term is the minimal error achievable by $\mathcal{H}$ with feature mapping $M_t$ on the target domain. When $\mathcal{H}$ is a complex family of hypothesis, such as neural networks, this term can be viewed as a lower bound of the adaptation performance, and it is determined by how difficult the learning problem on the target domain is.

Therefore, the only term we have control over is the discrepancy term, and thus by minimizing the discrepancy during training, we are reducing the upper bound on the adaptation performance. This validates the use of discrepancy in ADDA.

## E  Connection Between DGAN and Maxent

Both DGAN and maximum entropy (Maxent) are methods for density estimation. In this section we show that Maxent is a regularized version of DGAN.

Let $\Delta$ denote the simplex of all probability distributions over $\mathcal{X}$, and let $\mathbf{\Phi} : \mathcal{X} \to \mathbb{R}^d$ be the feature mapping. The maximum entropy (Maxent) model for density estimation solves the following optimization problem:

$$\max_{\mathbb{P} \in \Delta} \mathtt{H}(\mathbb{P}), \quad \text{s.t.} \left\| \mathbb{E}_{x \sim \mathbb{P}}[\mathbf{\Phi}(x)] - \mathbb{E}_{x \sim \widehat{\mathbb{P}}_r}[\mathbf{\Phi}(x)] \right\|_{\infty} \le \lambda,$$

where $\mathbf{\Phi} = (f_1, f_2, \ldots, f_n)$, and $\mathcal{F} = \{f_i, i \in [n]\}$ is the set of feature functions, $f_i \colon \mathcal{X} \to \mathbb{R}$. Note that $\max_{\mathbb{P} \in \Delta} \mathtt{H}(\mathbb{P})$ is equivalent to $\max_{\mathbb{P} \in \Delta} \mathrm{KL}(\mathbb{P} \parallel \widehat{\mathbb{P}}_r)$.

To see the connection between Maxent and DGAN, we can set $\mathcal{F} = \{\ell_{h,h'} \colon \ell_{h,h'}(x) = \ell(h(x), h'(x)), h, h' \in \mathcal{H}\}$, where $\mathcal{H}$ is the hypothesis set that defines the discrepancy $\mathrm{disc}_{\mathcal{H}, \ell}$. Then the Maxent optimization problem becomes

$$\max_{\mathbb{P} \in \Delta} \mathrm{KL}(\mathbb{P} \parallel \widehat{\mathbb{P}}_r), \quad \text{s.t.} \max_{h,h' \in \mathcal{H}} \left| \mathbb{E}_{x \sim \mathbb{P}} \left[ \ell\big(h(x), h'(x)\big) \right] - \mathbb{E}_{x \sim \widehat{\mathbb{P}}_r} \left[ \ell\big(h(x), h'(x)\big) \right] \right| \le \lambda. \tag{10}$$

In fact, we can write the dual problem of (10) as

$$\min_{\mathbb{P} \in \Delta} -\mathrm{KL}(\mathbb{P} \parallel \widehat{\mathbb{P}}_r) + \alpha \left\{ \max_{h,h' \in \mathcal{H}} \left| \mathbb{E}_{x \sim \mathbb{P}} \left[ \ell\big(h(x), h'(x)\big) \right] - \mathbb{E}_{x \sim \widehat{\mathbb{P}}_r} \left[ \ell\big(h(x), h'(x)\big) \right] \right| \right\}, \tag{11}$$

where $\alpha \ge 0$ is the Lagrange multiplier.

Recall that DGAN solves the following optimization problem:

$$\min_{\mathbb{P} \in \{\mathbb{P}_\theta : \theta \in \Theta\}} \max_{h,h' \in \mathcal{H}} \left| \mathbb{E}_{x \sim \mathbb{P}} \left[ \ell\big(h(x), h'(x)\big] \right) - \mathbb{E}_{x \sim \widehat{\mathbb{P}}_r} \left[ \ell\big(h(x), h'(x)\big) \right] \right|, \tag{12}$$

where $\{\mathbb{P}_\theta : \theta \in \Theta\}$ is a parametric family of distribution, $\{\mathbb{P}_\theta : \theta \in \Theta\} \subseteq \Delta$. Thus, the dual problem of Maxent (11) can be viewed as DGAN (12), plus a regularization term in the form of KL divergence $\mathrm{KL}(\mathbb{P} \parallel \widehat{\mathbb{P}}_r)$.

However, to use (11) under the DGAN framework, $\mathbb{P}$ is optimized over the special parametric family of distributions $\{\mathbb{P}_\theta : \theta \in \Theta\}$. The probability density of $\mathbb{P}_\theta(x)$ at any $x \in \mathcal{X}$ is unavailable, and thus we cannot directly compute the KL divergence $\mathrm{KL}(\mathbb{P}_\theta \parallel \widehat{\mathbb{P}}_r)$. One option is to use the Donsker-Varadhan [Donsker and Varadhan, 1975] representation:

$$\mathrm{KL}(\mathbb{P}_\theta \parallel \widehat{\mathbb{P}}_r) = \sup_{f \colon \mathcal{X} \to \mathbb{R}} \mathbb{E}_{x \sim \mathbb{P}_\theta} [f(x)] - \log \left( \mathbb{E}_{x \sim \widehat{\mathbb{P}}_r} [e^{f(x)}] \right).$$

Putting everything together, we get the regularized DGAN formulation that is motivated by the Maxent model: for some $\alpha > 0$,

$$\min_{\theta} \left\{ \inf_{f:\,\mathcal{X} \to \mathbb{R}} \log \left( \mathbb{E}_{x \sim \widehat{\mathbb{P}}_r}[e^{f(x)}] - \mathbb{E}_{x \sim \mathbb{P}_{\theta}}[f(x)] \right) \right.$$
$$\left. + \alpha \left\{ \max_{h,h' \in \mathcal{H}} \left| \mathbb{E}_{x \sim \mathbb{P}} \left[ \ell\big(h(x), h'(x)\big) \right] - \mathbb{E}_{x \sim \widehat{\mathbb{P}}_r} \left[ \ell\big(h(x), h'(x)\big) \right] \right| \right\} . \tag{13}$$