[Reviews · NeurIPS 2019]

Reviewer 1



The paper is original. The authors, despite similarities to LSGAN (Mao et al., 2017) do provide generality over the technique and provide theoretical guarantees about this. Overall, the paper quality is Okay. While there was some work to defend their claims, they are not fully backed up. First, they use discrepancy as both a loss function and a measure of performance. Of course their model would perform better on that metric: it's been directly trained to do so. A loss function is supposed to be a stand-in approximation for the true task for the model (E.g. no one is optimizing models for the end goal of reducing BCE loss, but rather, increasing accuracy or F1-scores on a test set), comparing losses directly is not useful. A better result would show the model performing better on some other metric (e.g. average performance of identically initialized networks on a test set trained only using generated data). Also a concern: selection of the hypothesis sets for the approximation of discrepancy, as far as I can tell, is not addressed beyond general phrases like "the set of linear mappings on the embedded samples." (This is excepting, of course, the linear case where they show it to be twice the largest eigenvalue of M(theta)) There should theoretically be an infinite number of linear mappings on the embedded samples, so how do they subselect? That would be something I'd like seen in the paper, or, at least, made more clear. Other small issues with the paper include some grammatical mistakes.

Reviewer 2



This papers proposes to use a discrepancy measure where an hypothesis set and a loss function are part of the discrepancy definition. In this very well written paper, the Authors make a compelling case for the potential benefits of a discrepancy measure capable of distinguishing distributions which would be considered as close by the Wasserstein Distance. The supplemental material is also a great addition to the flow of the paper allowing the Authors to clearly demonstrate their points. The DGAN derivations allow for a closed form solutions which can be generalized to a more complex hypothesis set if an embedding network is employed first. The paper is very convincing at presenting the potential advantages if one knows which loss and has a good enough set of hypotheses (from the embedding network). However, there is a bit of a sense of let down in the experimental section. The results for GAN training gets only a short treatment in 4.1 when it could have been a very strong point for the use of discrepancy. The results (7.02 IS, 32.7 FID) are reasonable but not a clear separator from other GAN methods. It would have been nice to have a thorough set of experiments there, instead of for ensembling. The EDGAN approach is working. However it is not as compelling a task as pure Generation. The model basically learns to ignore high discrepancy models and uses only output of a few low discrepancy models (as seen in weights in Table 4 in Supplemental) which is pretty much what intuition would dictate if you hand-picked the interpolation weights. You basically estimate your whole technique based on picking only 5 linear weights... If ensembling is the key to comparing your technique, some other methods out there (wasserstein model ensembling in ICLR'19) are also showing some decent results beyond model averaging. Overall a very good paper, experimental results could have focused more on DGAN than EDGAN. Note: line 205, should 'H' be the same font of the other hypothesis sets elsewhere in the paper? --------------- Thanks for the authors' rebuttal. I am maintaining my score of 7.

Reviewer 3



This paper considers learning generative adversarial networks (GAN) with the proposed generalized discrepancy between the data distribution P and generative model distribution Q. The discrepancy is novel in the sense that it takes the hypothesis set and the loss function into consideration. The proposed discrepancy subsumes Wasserstein distance and MMD as a special case by setting a proper class of hypothesis set and loss function. The author then proposed DGAN and EDGAN algorithms where they consider the set of linear functions with a bounded norm and loss function using square loss. Due to the restricted class of hypothesis set, the discrepancy has a closed-form simple estimation, which is very similar to matching covariance matrix of the data. For the EDGAN algorithm, they propose to learn the combination weights of a mixture of pretrained generators, which is a simple convex optimization problem. In short, the overall writing is clear and the setup is novel with the solid theoretical analysis.

[Author Response · NeurIPS 2019]

We thank all reviewers for their comments. Below, we respond to the questions raised by each reviewer.

## Reviewer #1

**Performance measure**: for DGAN trained on CIFAR 10, we reported the widely used Inception score (IS) and Fréchet
Inception distance (FID) (line 276 - 279), and our results are close to those of unconditional DCGANs. For EDGAN
on the toy example, we reported the likelihood of generated samples under the true model and vice-versa, which is a
more straightforward performance measure for generative models. Our EDGAN outperformed AdaGan on the two
likelihood-based measures.

For EDGAN on real-world examples, though, we cannot directly compute the likelihood, and it is unclear how to
evaluate the learned mixture weights. We will also report IS and FID scores in the final version. In fact, depending on
the choice of embedding networks, IS and FID scores can be very favorable. However, we do not believe that either IS
or FID fully captures the key objective of mixing GANs. We like the suggestion of the reviewer to report performance
results for a subsequent task based on generated data and will do so. We expect our DGAN algorithms to achieve a
high performance according to this metric since the discrepancy measure used for their training is precisely meant to
capture that. Note, however, if we only reported such results, others would argue that we had to report IS scores or
discrepancies, since that is what we optimize.

**Selection of the hypothesis set**: with a fixed embedding, the hypothesis set $\mathcal{H}$ used in experiments is always the set of
linear mappings with bounded $\ell_2$ norm, thus there is no further "selection of hypothesis set". Even though there are
infinitely many linear mappings in $\mathcal{H}$, discrepancy takes a supremum over them (see Eq. 1) and thus is well defined.
Proposition 3 actually gives a closed-form expression for the discrepancy when $\mathcal{H}$ is the set of linear mappings. We
adopted linear functions and the squared loss in our experiments since the discrepancy admits a closed-form solution in
that case. However, the learner could choose other hypothesis sets $\mathcal{H}$ and loss functions $\ell$ relevant to the learning tasks
and our theoretical results would still apply.

## Reviewer #2

**DGAN experiments**: we will include more experimental results for DGAN on more datasets and report their IS and
FID scores.

**Other ensemble methods**: the ICLR'19 work is interesting and we could potentially use discrepancy instead of the
Wasserstein distance there to come up with new ensemble algorithms.

## Reviewer #3

**Compare with McGan**: indeed, in the specific case where $\mathcal{H}$ is the family of bounded norm linear functions and the
loss is the squared loss, DGAN coincides with one of the objectives sought by McGan, that of matching the empirical
covariance matrices of the two distributions, however the norm used in McGan is different (nuclear norm). Nevertheless,
our theoretical analysis can serve as a justification of McGan in that case. We thank the reviewer for the connection and
will reference the paper. In general though, for other hypothesis sets and loss functions, the objectives and techniques
are distinct.

**Fair comparison**: in the DGAN experiments, we used the DCGAN architecture and trained our own embedding layer
(line 258 - 261). We used pre-trained embedding only in EDGAN experiments, where the goal is to mix pre-trained
GANs, and thus it is less *unfair* to assume a pre-trained embedding.

**Ensemble weights**: we reported the learned ensemble weights in Table 4 in Appendix C. In most cases, the weights are
not very sparse and at least two GANs are assigned non-zero weights.

[Meta-Review · NeurIPS 2019]

The reviewers agree that this work is a useful contribution to the literature. Please take in to account the reviewer suggestions in the final camera-ready version.